# Distinct neuropeptide-receptor modules regulate a sex-specific behavioral response to a pheromone

Douglas K. Reilly[1,5], Emily J. McGlame [1,6], Elke Vandewyer[2], Annalise N. Robidoux[1], Caroline S. Muirhead[1], Haylea T. Northcott[1,7], William Joyce[3], Mark J. Alkema[3], Robert J. Gegear[4], Isabel Beets [2] & Jagan Srinivasan [1✉]

Dioecious species are a hallmark of the animal kingdom, with opposing sexes responding differently to identical sensory cues. Here, we study the response of *C. elegans* to the small-molecule pheromone, ascr#8, which elicits opposing behavioral valences in each sex. We identify a novel neuropeptide-neuropeptide receptor (NP/NPR) module that is active in males, but not in hermaphrodites. Using a novel paradigm of neuropeptide rescue that we established, we leverage bacterial expression of individual peptides to rescue the sex-specific response to ascr#8. Concurrent biochemical studies confirmed individual FLP-3 peptides differentially activate two divergent receptors, NPR-10 and FRPR-16. Interestingly, the two of the peptides that rescued behavior in our feeding paradigm are related through a conserved threonine, suggesting that a specific NP/NPR combination sets a male state, driving the correct behavioral valence of the ascr#8 response. Receptor expression within pre-motor neurons reveals novel coordination of male-specific and core locomotory circuitries.

[1] Department of Biology and Biotechnology, Worcester Polytechnic Institute, Worcester, MA, USA. [2] Neural Signaling and Circuit Plasticity Group, Department of Biology, KU Leuven, Leuven, Belgium. [3] Neurobiology Department, University of Massachusetts Medical School, Worcester, MA, USA. [4] Department of Biology, University of Massachusetts Dartmouth, Dartmouth, MA, USA. [5]Present address: Tufts University, Medford, MA, USA. [6]Present address: AbbVie Foundational Neuroscience Center, Cambridge, MA, USA. [7]Present address: Optum, Hartford, CT, USA. ✉email: jsrinivasan@wpi.edu

Sex-specific behaviors are unique aspects of survival throughout the animal kingdom from invertebrates to humans[1–3]. These behaviors include a wide range of coordinated and genetically preprogrammed social and sexual displays that ensure successful reproductive strategies, ultimately resulting in survival of the species in its natural environment[2,3]. The neural circuits regulating these behavioral responses are conserved, and often shared between sexes, but dependent on social experience and physiological state[2]. For example, the vomeronasal and main olfactory epithelium in mice are required for male aggression and mating, but in females they contribute towards receptivity and aggression[3]. Prominent among these stimuli are mating cues[4]; while the visual displays of higher order animals are among the most apparent of these, chemical mating cues are the most ubiquitous, with entire sensory organs dedicated to pheromone sensation[5].

Pheromones are small-molecule signals between conspecifics that convey information on the sender's current physiological state, and potentially life stage and developmental history[6–8]. How the nervous system responds to these stimuli is dependent on both the internal, physiological state of the animal[9], and external, concurrently sensed stimuli[10]. We have previously shown that behavioral response to pheromones is directly dependent on the physiological state of the animal[11], though the mechanisms which determine such responses remain enigmatic.

Nematodes communicate through a large and growing class of pheromones termed ascarosides (ascr)[6]. These small molecules convey social as well as developmental information, and the assays used to understand the roles of these cues have varied[4,6]. There are multiple ascarosides found to communicate attractive behaviors, specifically in a sex-specific manner, including: ascr#1, ascr#2, ascr#3, ascr#4, and ascr#8[12]. Unique among ascaroside structures is the presence of a *p*-aminobenzoate group—a folate precursor that *C. elegans* are unable to synthesize, yet obtain from bacterial food sources[13]—at the terminus of ascr#8. This pheromone has previously been shown to act as an extremely potent male attractant, being sensed via a chemosensory pathway shared with ascr#3: the male-specific CEM neurons[14]. However, whereas ascr#3 is also sensed by over half a dozen chemosensory neurons[14–17], ascr#8 has only been shown to be sensed by the male-specific CEM[14].

While the CEMs offer a sex-specific mode of chemosensation for ascr#8, neuromodulators and hormones are heavily implicated in all stages of the of sex-specific and pheromone-elicited *C. elegans* mating behaviors. Prior to sensation of mating cues, the mate searching behavior of male *C. elegans* is modulated by the neuropeptide, PDF-1[18,19]. Interestingly, this neuropeptide also controls the sexual identity of the ASJ chemosensory neurons[20]. The mating pheromone ascr#3 is modulated by insulin signaling[21], while activation of ascr#3-sensing neurons also activates the NPR-1 receptor[17]. Finally, the physical act of male sexual turning during mating is mediated by multiple FMRFamide-like peptides[22]. This complex regulation of behaviors relies on specific neuropeptide–neuropeptide receptor (NP/NPR) modules.

Unique NP/NPR modules are known to drive specific physiological and behavioral responses in *C. elegans*[23]. While DAF-2 propagates insulin-like peptide (*ins*) signaling, the specific peptide determines the effect. For example, *ins-4* functions in learning, while *ins-6* affects synapse formation[24,25]. Meanwhile, avoidance of ascr#3 by hermaphrodites is mediated in part by INS-18/DAF-2 signaling—higher levels of *ins-18* expression result in lower ascr#3 avoidance rates[21]. Conversely, FMRFamide-like peptide (*flp*) genes, many of which encode multiple peptides, signal through a complex network in which multiple receptors sense identical peptides, and multiple FLP peptides activate the same receptors. For instance, activation of NPR-4 by FLP-18 modulates reversal length[26], while the sensation of the divergent FLP-4 by the same receptor contributes to food preference choice[27].

Here, we investigate the neuronal mechanisms governing the behavioral attractive response of male *C. elegans* to ascr#8[28]. Males exhibit a unique behavioral tuning curve to ascr#8, preferring concentrations in the 1 µM range, no longer being attracted to higher concentrations[14]. Given that multiple *flp* NP/ NPR modules have been shown to play roles in setting physiological state[29], as well as linking sensation to physiology and behavior[30–32], we reasoned that peptidergic signaling is likely to play a role in the male ascr#8 behavioral response.

Previous studies of *C. elegans* behavioral responses to attractive social ascarosides employed a Spot Retention Assay (SRA)[14,33]. However, we found that the SRA contains several drawbacks, including male–male contact and the inability to track individual animals through the course of an assay. To address these issues, we have developed a single worm attraction assay (SWAA): a more robust assay that determines variables on a per-worm basis, and not solely at the population level. We utilized our novel SWAA to examine the responses of *him-8* males defective in *flp* neuropeptide genes expressed in male-specific neurons; *flp-3, flp-6, flp-12,* and *flp-19*[34]. In doing so, we discovered that *flp-3* plays a role in determining the sex-specific behavioral valence: i.e., determining whether the response to ascr#8 is attractive or aversive[16,35,36].

We identified two divergent FLP-3 receptors responsible for sensing the processed neuropeptides. Receptor activation studies elucidated that the previously identified *flp-3*-sensing G protein-coupled receptor, NPR-10[23], and the novel FRPR-16, are both activated by FLP-3 peptides at nanomolar affinities. Additionally, loss-of-function mutations in either receptor result in behavioral defects that parallel those observed in *flp-3* mutants.

To more completely understand the role of *flp-3* in mediating the ascr#8 behavioral response, we adapted a peptide rescue-by-feeding protocol[37]. Using this method, we were able to rescue individual peptides in *flp-3* mutant animals and showed that a specific subset of FLP-3 peptides responsible for suppressing the avoidance differs from those responsible for driving male attraction to ascr#8.

Here we show that individual neuropeptides encoded by the *flp-3* gene exhibit specific biological activity, by binding multiple receptors, to drive the behavioral valence to a cue in a sex-specific manner.

## Results

**Spot retention assay vs. single worm assay**. We adapted the spot retention assay (Supplementary Fig. 1)[28,38] to allow for better characterization of individual worm behavior and robust interrogation of attraction to small molecules. We first compared the attractiveness of 1 µM ascr#8 (Fig. 1a, inset) across multiple strains of *C. elegans* (the wild-type N2 strain, the high incidence of male *him-5* and *him-8* strains, and the chemosensory cilia defective *osm-3*), using our novel behavioral assay, the single worm attraction assay (SWAA) (Fig. 1a). In this assay, individual animals are placed directly into the spot of the ascaroside cue (A) while simultaneously removing any potential of male–male contact. A spatial control is included throughout the assay plate to allow us to investigate any innate differences in the number of visits to the well center, or the time spent therein, of which we have only found one strain to date with differences in male dwell time (Fig. 1a). Likewise, a vehicle control (V) is also included, to account of any changes in dwell time that are driven by the components of the vehicle.

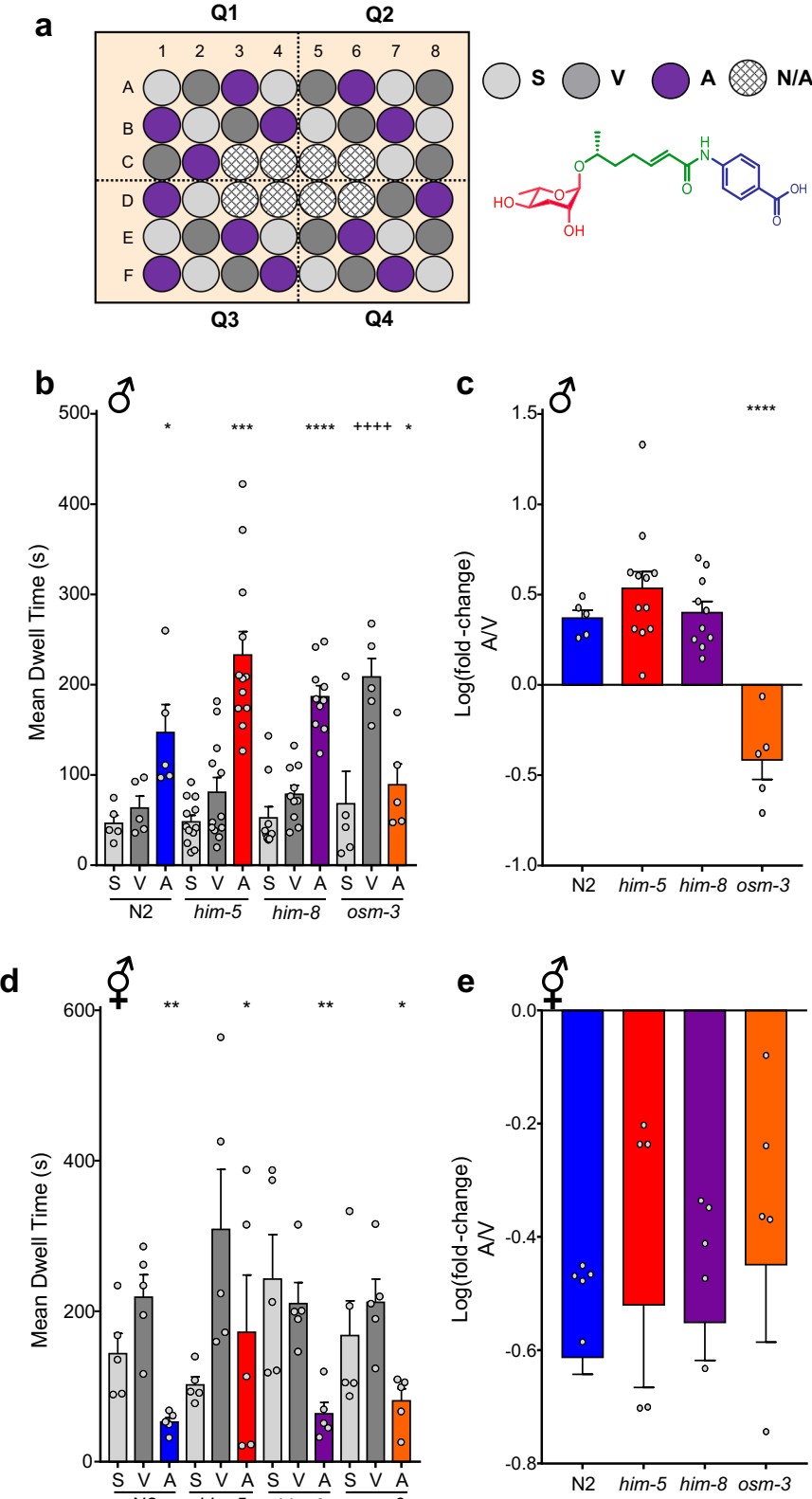

**Fig. 1 Attraction to ascr#8 is sex specific. a** Design of the Single Worm Attraction Assay (SWAA). The outer 40 wells of a 48-well suspension cell culture plate are seeded with NGM agar and a thin lawn of OP50 *E. coli*. A random block design results in spatial control (light gray), vehicle control (dark gray), and ascaroside (purple) containing wells. Quadrants are recorded for 15 m. (Inset) The structure of ascr#8. **b** Raw dwell times of males of N2 (blue), *him-5* (red), *him-8* (purple), *osm-3;him-5* (orange) in SWAA. **c** Transformed log(fold-change) of male dwell time data. **d** Raw dwell time of hermaphrodites across strains and **e** log(fold-change) of SWAA data of hermaphrodites. Light gray denotes spatial controls ("S") (when applicable), dark grey denotes vehicle controls ("V"), colors denote ascr#8 values ("A"). ♂ denotes male data, ⚥ denotes hermaphrodite data. For all figures: Error bars denote SEM. $n \geq 5$. *$p < 0.05$, **$p < 0.01$, ***$p < 0.001$, ****$p < 0.0001$, unless denoted otherwise. For **1b**: ++++$p < 0.0001$, vehicle vs. spatial control.

Male *C. elegans* exhibited a significant increase in the amount of time spent within ascr#8 spot compared to the vehicle control, in all wild-type and *him* male strains tested (Fig. 1b, c). These results confirm the attraction behavior of wild-type and *him* males observed with SRA (Supplementary Fig. 1). To measure the attraction behavior of the ascaroside, we then computed the normalized increase in dwell time, calculated as log(fold-change) [i.e., ascaroside dwell time over vehicle dwell time]. Using this log(fold-change) metric, we observed that the increase in attraction to ascr#8 is consistent across N2, *him-5*, and *him-8* male strains (Fig. 1c). This metric enabled a direct comparison between strains and conditions while accounting for baseline variability in vehicle dwell times.

Given the setup of the SWAA assay, we can measure the number of visits per-worm under different genetic backgrounds and conditions (Supplementary Fig. 2a, c). This assay additionally helps calculate the percentage of attractive visits to the cue (Supplementary Fig. 2b). Our results suggest no difference in the attractive properties to ascr#8 between any of the wild-type strains (Supplementary Fig. 2b). Setting an "attractive visit" as any visit longer than two standard deviations above the mean vehicle dwell time, we show that males are indeed attracted to the cue itself, and not the male–male contact. Our results suggest that it is in fact a minority of animals (30–45%) that exhibit attractive visits to the cue (Supplementary Fig. 2b). This rate of behavioral attraction to ascr#8 is consistent with calcium imaging experiments wherein ascr#8 exposure elicits similar rates of calcium transients in the CEM neurons[39].

Unlike the SRA, wherein hermaphrodites did not exhibit any difference in dwell time between vehicle and ascr#8 (Supplementary Fig. 2d, e), the SWAA revealed that hermaphrodites from all strains consistently spent significantly less time in ascr#8 than the vehicle, with no difference between the spatial and vehicle control dwell times (Fig. 2d, e). Hermaphrodites also visited the ascaroside cue less than they did vehicle or spatial control well centers and exhibited little-to-no attractive visits (Supplementary Fig. 2c, d).

Together, these data validate the SWAA as a robust assay for the measurement of the attractiveness of a small-molecule cue on a single animal basis in both sexes. In addition, it provides data on visit count and the percent of attractive visits that was previously impossible utilizing the SRA. Interestingly, attractive visits to the ascr#8 cue were only observed in 30–45% of the time (Supplementary Fig. 2b), suggesting that the individual state of the animal plays a critical role in determining the behavioral response to the ascaroside, as seen in other ascaroside behavioral responses[11,40].

**Peptidergic signaling drives sex-specific ascr#8 behavioral response.** Several neuropeptides of the FMRFamide-like-peptide (FLP) family have been implicated in the mechanosensory regulation of male-mating behavior[37]. The genes encoding the neuropeptides *flp-8*, *flp-10*, *flp-12*, and *flp-20* all suppress the number of turns around a hermaphrodite executed by a male prior to mating[22]. Despite this enrichment of *flp* genes functioning in the mechanosensation of these male-specific behaviors[22], there has been no neuropeptide found to regulate the chemosensation of mating ascarosides. We sought to understand why an attractive concentration of a mating pheromone does not result in consistent attraction (Supplementary Fig. 2b) by investigating potential peptidergic signaling pathways that function in the sensation of ascr#8.

We focused our initial screening of neuropeptides on the FLP family. We generated *him-8* lines of *flp* genes expressed in male-specific neurons, specifically *flp-3*, *flp-6*, *flp-12*, and *flp-19*[34]. To

avoid confounding variables, our criteria for selection stipulated that outside of male-specific neurons, expression profiles would be limited to a small number of neurons (*flp-5* was therefore excluded as it exhibits expression in the pharyngeal muscle; while *flp-21* and *flp-22* are expressed in a large number of neurons outside of the male-specific expression profiles).

We found that loss of *flp-3* strongly affected the ability of male *C. elegans* to respond to ascr#8, (Fig. 2a, b, Supplementary Fig. 3a, b). The log(fold-change) of *flp-3* is the only value significantly different than that seen in the wild-type (Fig. 2b). Interestingly, there was defect seen neither in *flp-3* hermaphrodites, nor any other strain (Fig. 2d, e, Supplementary Fig. 3c, d).

Because the defect in male response to ascr#8 was significant, and the SWAA was designed to detect attractive behaviors, we sought to determine if *flp-3* loss-of-function (*lof*) animals were in fact avoiding ascr#8. Using a previously described drop avoidance assay[11,41], we exposed forward moving animals to a drop of either vehicle control or ascr#8 and scored the avoidance index. Wild-type males did not avoid the cue, as expected for an attractive cue, while *flp-3 lof* males strongly avoided the pheromone (Fig. 2c). The hermaphroditic behavior was unaffected by the loss of *flp-3*. (Fig. 2f). Together, these results suggest that *flp-3* functions to control the behavioral valence of the ascr#8 response to be attractive in a sex-specific manner; serving in males to suppress a basal avoidance behavior observed in hermaphrodites.

Rescue of *flp-3* under a 4-kb region of its endogenous promoter was able to restore the behavioral valence of males to wild-type levels (Fig. 2g–i, Supplementary Fig. 3e, f). While overexpression of neuropeptides can result in dominant negative phenotypes[18], expression of the *flp-3* construct in wild-type animals did not alter wild-type behavioral response to ascr#8 (Fig. 2g–i, Supplementary Fig. 3e, f).

To rule out an allele specific effect of the *flp-3(pk361)* mutation, which results in deletion of the entire coding sequence as well as 439 bp of upstream and 1493 bp of downstream genomic sequence[42], we also assayed *flp-3(ok3265)*, an in-frame deletion of the coding sequence that retains expression of two peptides produced by the *flp-3* gene[43] (FLP-3-1 and FLP-3-4) (Supplementary Fig. 4a). The *flp-3(pk361)* and *flp-3(ok3265)* mutant phenotypes were identical (Supplementary Fig. 4b–f), confirming that the deletion in the *pk361* allele did not cause any off-target effects, and that neither of the two peptides still encoded by the *ok3625* allele (FLP-3-1 and FLP-3-4) are sufficient to rescue the mutant phenotype. Whether these peptides contribute to the ascr#8 response in an another way remains to be elucidated.

**FLP-3 functions specifically to modulate the ascr#8 behavioral response.** While ascr#8 is a potent male-attracting pheromone, previous studies have shown that ascr#2, ascr#3, ascr#4 also function synergistically in attracting males[33]. The CEM neurons that are required for ascr#8 sensation also function in ascr#3 sensation[14]. While ascr#3 signal propagation is processed through the hub-and-spoke circuit centered around RMG[16,17,44], little is known about the mechanics of ascr#8 sensation outside of CEM involvement[14]. To determine if *flp-3* functions to regulate pheromone-mediated male attraction and avoidance in a general manner, or rather one specific to ascr#8, we assayed the response of wild-type and *flp-3 lof* males to ascr#3, a cue for which behavioral valence has also recently been shown to be regulated in a sex-specific manner[16]. We found that *flp-3 lof* males exhibited no defect in their attractive response to ascr#3 (Supplementary Fig. 5), suggesting that its role is indeed specific to that of ascr#8 sensation.

Expression analysis of a FLP-3 translational fusion (p*flp-3*::*flp-3*::mCherry) confirmed previous expression analyses of the

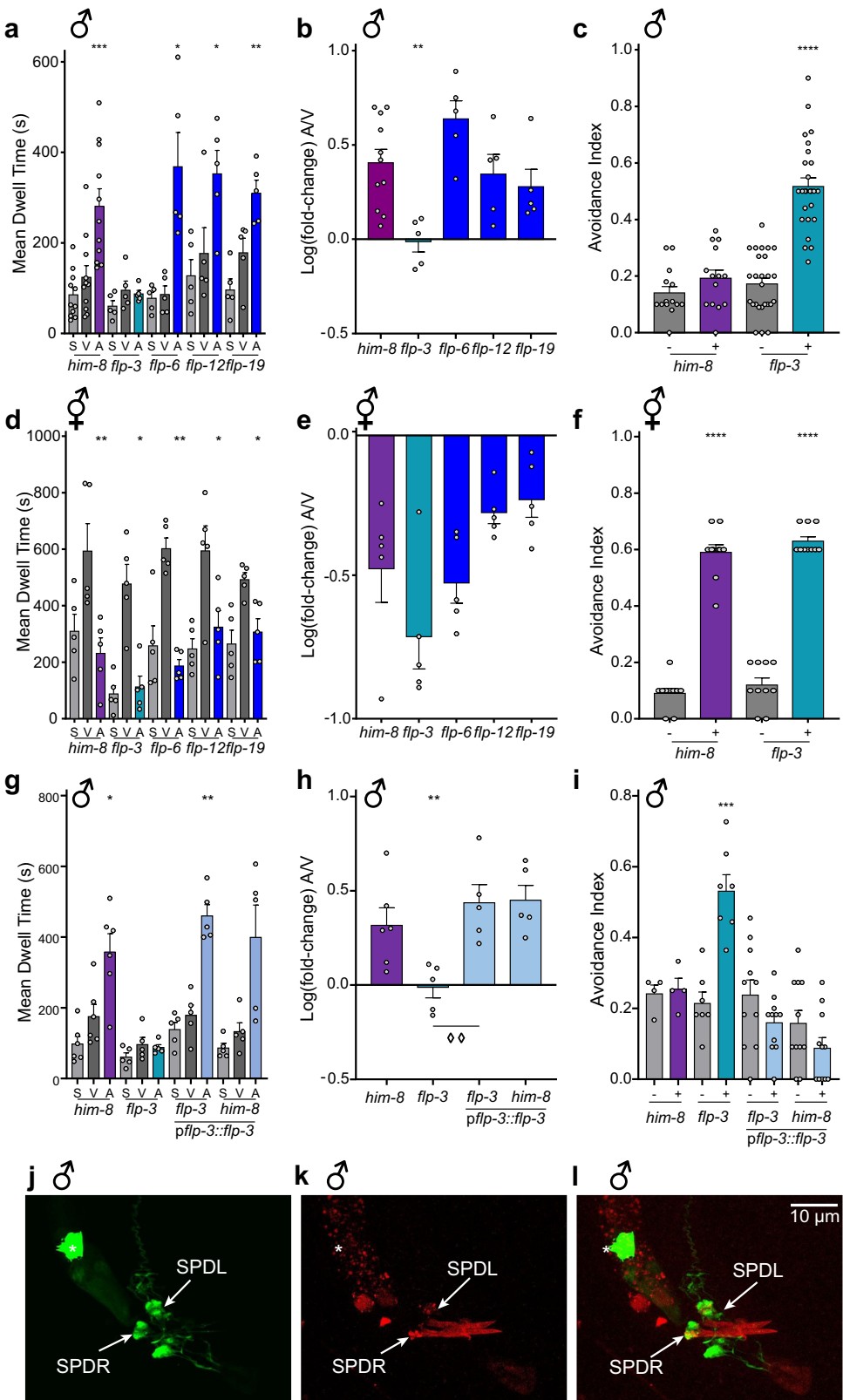

neuropeptide within male-specific spicule neurons[34,45] (Fig. 2j–l, Supplementary Fig. 6a–c). Transcriptional reporters have shown robust *flp-3* expression in the amphid IL1 neurons, as well as the sensory PQR and the male-specific interneuron, CP9, although our translational fusion exhibited no PQR or CP9 expression (Fig. 2j–l, Supplementary Fig. 6a–c). Previous studies employed 1–2 kb regions of promoter sequence driving GFP expression,

while our construct employs a 4 kb region, thereby incorporating further regulatory elements that may restrict expression patterns. By including the full coding sequence in our translational fusion, we have also incorporated the regulatory elements found within the introns of the *flp-3* gene[46].

Interestingly, we observed localization of mCherry within sensory cilia of male dorsal and ventral IL1 neurons, as well as

**Fig. 2 Peptidergic regulation of the behavioral response to ascr#8 is sex specific.** Screen of FMRFamide-like peptide (*flp*) defective mutants in response to ascr#8 in SWAA. **a** Male raw dwell time and **b** log(fold-change) values revealed specificity of *flp-3* in the ascr#8 behavioral response. **c** A drop avoidance assay was employed to reveal a change in behavioral valence, with *flp-3* males avoiding ascr#8 (See *Methods* for details). **d** Hermaphrodite raw dwell times and **e** log(fold-change) values in response to ascr#8 are consistent across *flp* mutants. **f** ascr#8 avoidance in hermaphrodites is unaffected by the loss of *flp-3*. **g, i** Attractive behavior in response to ascr#8 is restored in *flp-3* mutant males by expressing *flp-3* under its endogenous promoter. **g** raw dwell times in males and **h** log(fold-change) values), as well as **i** rescue of *flp-3* restores avoidance to ascr#8. **j–l** Expression pattern of p*flp-3*::*flp-3*::mCherry in the male tail. Colocalization with *gpa-1*::GFP in the SPD spicule neurons suggests that *flp-3* is expressed in male-specific neurons in the tail (arrows). Asterisk denotes coelomocyte accumulation of GFP. **j** GFP, **k** mCherry, **l** merged image at ~×90 magnification. In panel **h**, ◇◇$p < 0.01$ for *flp-3* mutant versus transgenic rescue. Light gray denotes spatial controls ("S") (when applicable), dark gray denotes vehicle controls ("V"), colors denote ascr#8 values ("A") (*him-8*, purple; *flp-3*, teal; *flp* mutants, blue; transgenic rescues, light blue). ♂ denotes male data, ⚥ denotes hermaphrodite data.

in puncta spanning their dendrites (Supplementary Fig. 6d, e), consistent with peptide packing into dense core vesicles[47]. Previous studies have observed dense core vesicles within, and being released from, dendritic arbors[48], supporting our findings that FLP-3 is packaged within the dendrites of the IL1 neurons (Supplementary Fig. 6a–c). We found an identical expression pattern within the IL1 cilia and dendritic puncta of hermaphrodites as well (Supplementary Fig. 6f). Recent single-cell RNA-sequencing of the adult nervous system has again found more prolific expression of *flp-3* throughout the nervous system, including most of the VC neurons[49]. However, these studies were performed only in hermaphrodites, and were therefore unable to examine any male-specific changes in expression. In males, our *flp-3* construct also exhibited male-specific tail expression in the SPD spicule neurons (Fig. 2j–l, Supplementary Fig. 6a–c), and coupled with IL1 expression, completely rescued the attractive response to ascr#8 (Fig. 2g–i, Supplementary Fig. 3e, f), suggesting a physiologically relevant site-of-release from this small subset of neurons.

Because the spicule neurons are exposed to the environment[50], we investigated whether they play a direct role in the sensation of ascr#8. To test this, we assayed *ceh-30 lof* males for their ability to avoid ascr#8. Male *ceh-30 lof* animals lack the male-specific CEM neurons responsible for ascr#8 sensation in the amphid region of the animal[14,51]. *him-5* males did not avoid ascr#8 (Supplementary Fig. 7). Males lacking CEM neurons also did not avoid ascr#8 (Supplementary Fig. 7). However, with *flp-3* still present in these animals, it may be that they are still able to sense the cue, but do not avoid it due to the presence of the neuropeptide. We therefore generated a *ceh-30;flp-3* double mutant, and found that these animals still do not avoid the pheromone (Supplementary Fig. 7), confirming that the CEM neurons are the primary route of ascr#8 chemosensation which results in the male *C. elegans* behavioral response[14].

**FLP-3 regulates attractive behavior to ascr#8 by activation of two evolutionarily divergent G protein-coupled receptors.** The *flp-3* gene encodes multiple peptides[23]. Recent studies have uncovered a tenth peptide encoded by the gene; although this newest peptide does not contain the conserved GTMRFamide motif found in the remainder of *flp-3* peptides (Fig. 3a)[52]. We determined that the lysine-arginine sites flanking the individual peptides are processed specifically by the proprotein convertase encoded by the *egl-3* gene, and not by *aex-5* or *bli-4*[22,53,54] (Supplementary Fig. 8), supporting previous studies that the *egl-3* gene is the proprotein convertase involved in mating behaviors[22].

To better understand where the fully processed peptides act within the male-specific circuit, we assayed mutants for receptors that have previously shown activation upon FLP-3 peptide exposure. While activation of NPR-4 has been reported for only two peptides encoded by *flp-3*, NPR-5 and NPR-10 have been shown to respond to four and six *flp-3* encoded peptides, respectively (Fig. 3a)[23]. Recent studies have considered NPR-4

and NPR-10 to be representative of one another due to their close phylogenetic relationship and separation by a recent gene duplication[55,56]. However, in our testing of these mutants using our SWAA, we found that *npr-4* and *npr-5 lof* males respond similarly to *him-8* males (Fig. 3b–d, Supplementary Fig. 9) while *npr-10 lof* animals exhibited a complete loss of attraction to the cue, as well as a partial avoidance phenotype matching that of *flp-3 lof* mutants (Fig. 3b–d, Supplementary Fig. 9).

Transgenic rescue by an NPR-10::GFP translational fusion construct expressed under 1.6 kb of the endogenous promoter was able to restore wild-type levels off attraction in an *npr-10 lof* mutant background (Fig. 3c, Supplementary Fig. 9a, b). This construct was also able to suppress the avoidance phenotype of *npr-10* (Fig. 3d, Supplementary Fig. 9c). Expression analysis of NPR-10::GFP revealed expression in both amphid and phasmid regions of the animals (Fig. 3e, f. Supplementary Fig. 9d, e). Among these head neurons are the inner labial IL2 neurons, as well as their respective socket cells (ILso). NPR-10::GFP fluorescence was also observed in the ADL and ASG chemosensory neurons, cells which synapse onto AVD and AIA neurons, contributing to reversal control and turning circuitries, respectively. The localization of NPR-10 in the RMEL and RMEV neurons provides a direct input into neurons innervating muscle cells (Fig. 3e). Alongside expression in the interneuron AVK (Fig. 3e), which links the *npr-10* circuitry to the backwards locomotion neuron AVE, AVF expression (Fig. 3e) links the circuit to the forward locomotion premotor interneuron, AVB.

NPR-10 expression was also observed in the B-class ray neurons in the male tail (Fig. 3f). Given the tight localization of these cells, it may be that NPR-10 is also present in the HOB neuron, which is impossible to decipher without further colocalization studies. However, the RnB neurons that express NPR-10 to sense SPD-secreted FLP-3 peptides heavily innervate the male-specific interneuron, EF1, which travels from the tail to synapse onto neurons in the head of the animal, including the forward locomotion neuron, AVB. Interestingly, the hermaphrodite tail exhibits expression in the dorso-rectal ganglion neurons DVA, DVB, DVC, and ALN (Supplementary Fig. 9e), which is not observed in the male tail. These data suggest that NPR-10 plays sex-specific roles based on sex-specific expression patterns.

Using Chinese hamster ovarian (CHO) cell cultures stably expressing the promiscuous G protein, Gα16, and the calcium reporter, aequorin[57], we found that both isoforms of NPR-10 are activated by seven of the ten FLP-3 peptides (Supplementary Fig. 10a, b), with half-maximal effective concentrations (EC$_{50}$) in the nM range (Fig. 4a–i). Peptide FLP-3-6 (EDGNAPFGTMK-Famide) did not activate NPR-10 in our assay. This peptide contains an R-to-K mutation within the C-terminal motif, which may explain the lack of receptor activation. Likewise, peptide FLP-3-10 (STVDSSEPVIRDQ), which contains no sequence homology with any RFamide peptide (Fig. 4i) also failed to activate the receptor. Interestingly, FLP-3-8 (SADD-SAPFGTMRFamide) did not activate either NPR-10A or NPR-

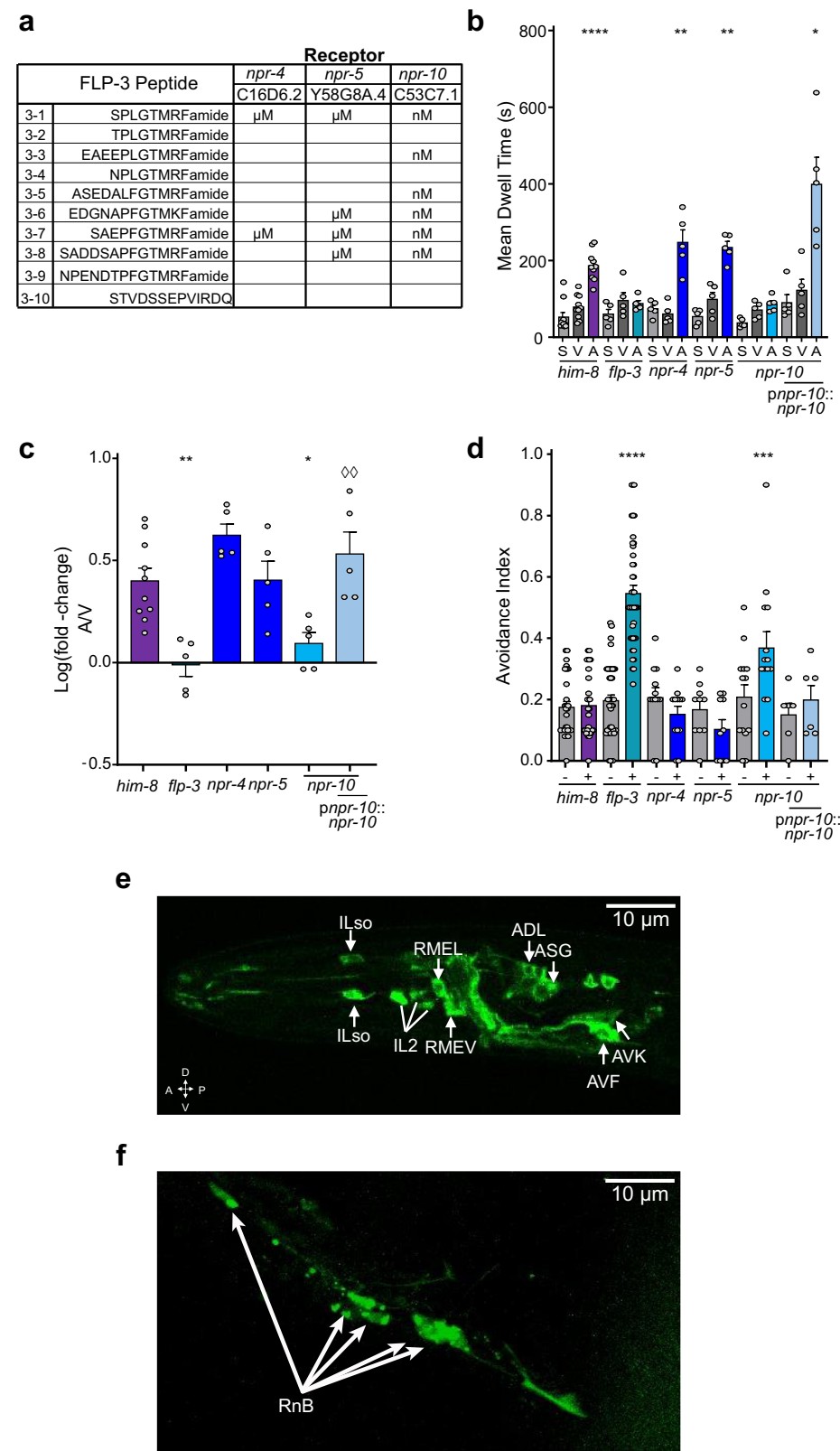

**Fig. 3 The G protein-coupled receptor, NPR-10, is required for the male behavioral response to ascr#8. a** FLP-3 peptide affinities for known receptors, adapted from Li and Kim, 2014[23]. **b, c** Male raw dwell time and log(fold-change) values for *npr* receptor mutants and *npr-10* rescue in the SWAA, respectively. **d** *npr-10* males exhibit increased avoidance to ascr#8 compared to the other receptors. Expression of *npr-10* under its endogenous promoter rescues the avoidance phenotype. **e** Localization of p*npr-10*::*npr-10*::GFP in the amphid region of the male head. Neurons expressing *npr-10* include the inner labial neurons, IL2, and their respective socket cells (ILso), the interneurons RMEV and RMEL, the chemosensory neurons ADL and ASG, as well as the interneurons AVF and AVK. **f** Expression of p*npr-10*::*npr-10*::GFP in the mail tail. Localization is observed in the B-class Ray neurons. ♂ denotes male data.

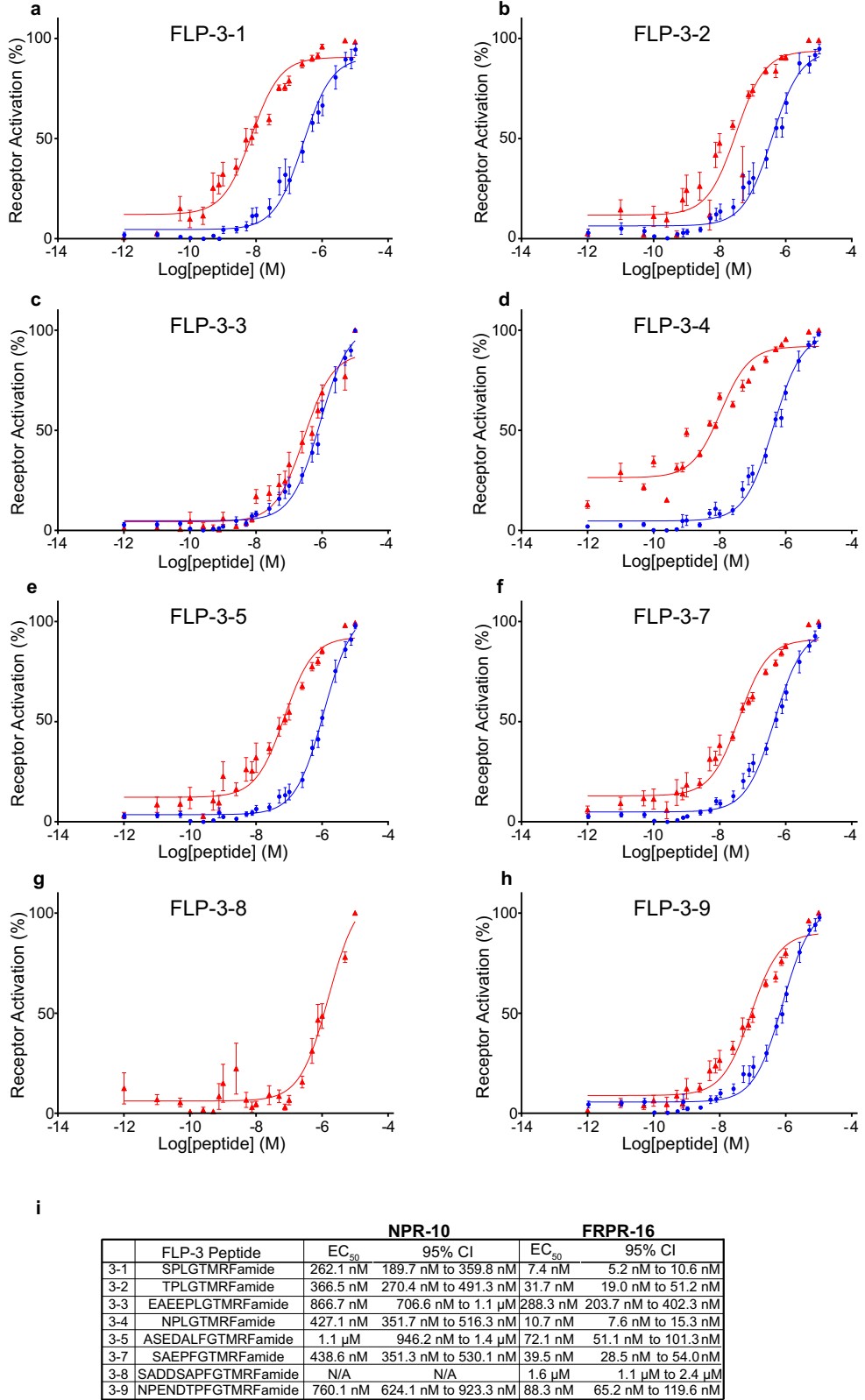

**Fig. 4 FLP-3 peptides activate two different G protein-coupled receptors, NPR-10 and FRPR-16, in vitro.** Dose response curves of **a** FLP-3-1, **b** FLP-3-2, **c** FLP-3-3, **d** FLP-3-4, **e** FLP-3-5, **f** FLP-3-7, **g** FLP-3-8, and **h** FLP-3-9 for activation of NPR-10B (blue circles) and FRPR-16 (red triangles). Peptides FLP-3-6 and FLP-3-10 did not activate either receptor. **i** $EC_{50}$ values and 95% Confidence Intervals for FLP-3 peptide activating NPR-10B and FRPR-16. **a–h** Error bars denote SEM. $n \geq 6$.

10B, despite its conserved terminal amino acid sequence (Supplementary Fig. 10a, b).

The lack of full avoidance phenotype observed in npr-10 lof mutants suggests that there are other FLP-3 receptors involved in regulating the ascr#8 avoidance behavior. Interestingly, while the human NPY/Drosophila NPF receptor NPR-10 is required for FLP-3 sensation, the Drosophila FR receptor homolog, FRPR-16, was also found to be reliably activated by FLP-3 peptides in vitro (Supplementary Fig. 10c). This evolutionarily divergent receptor exhibited potencies in the 10-nanomolar range for seven of the peptides, and submicromolar for an eighth peptide (Fig. 4). Again, FLP-3-6 and FLP-3-10 did not activate FRPR-16, supporting the notion that the terminal motif conserved in the remaining FLP-3 peptides is critical for receptor activation. Cells transfected with a control vector did not exhibit any activation following exposure to FLP-3 peptides, confirming that the activation observed is specific to receptor-ligand interactions with NPR-10 and FRPR-16 (Supplementary Fig. 10d).

A full-gene deletion of frpr-16 was generated using CRISPR mutagenesis (Fig. 5a, b)[58]. We assayed frpr-16 lof males for their ability to respond to ascr#8. Males lacking frpr-16 exhibited a loss of attraction to ascr#8, as well as a partial avoidance phenotype, like that observed in npr-10 lof mutant animals (Fig. 5, Supplementary Fig. 11). Interestingly, a double mutant containing both npr-10 and frpr-16 null alleles not only did not result in an additive effect in the avoidance phenotype, but in fact also suppressed the avoidance phenotype, suggesting that these receptors are nonredundant in their functions (Fig. 5e, Supplementary Fig. 11c).

Rescue of FRPR-16 under 1.9 kb of its endogenous promoter was able to restore wild-type attractive behavior (Fig. 5c, d, Supplementary Fig. 11a, b), as well suppress the avoidance phenotype (Fig. 5e, Supplementary Fig. 11c). Localization of the mCherry fusion reporter was observed in the premotor interneurons responsible for reverse locomotory control: AVA, AVE, and AVD (Fig. 5f). The fluorescent protein was also seen anterior to the nerve ring in the gas-sensing BAG neuron (Fig. 5f). This expressing pattern was not sex specific, as a matching expression pattern was observed in hermaphrodites (Supplementary Fig. 11d). Together, these data show that NPR-10 and FRPR-16 function as receptors for FLP-3 peptides.

**Rescue of individual FLP-3 peptides by feeding reveals a specific subset of active peptides required for attractive behavior.** To identify which FLP-3 peptides are required for male avoidance of ascr#8, we adopted a peptide feeding approach[59], similar to RNAi feeding, as initially described previously[37,59] (Fig. 6a). Using Gateway Cloning technology, we first inserted FLP-3 peptide coding sequences into a bacterial expression vector. As a control, we first tested the FLP-3 peptides that are still encoded in the ok3625 allele which are insufficient to drive the proper ascr#8 response in vivo: FLP-3-1 (SPLGTMRFamide) and FLP-3-4 (NPLGTMRFamide) (Supplementary Fig. 4). We then reared flp-3 lof animals on lawns of bacteria expressing the rescue constructs, and their progeny were assayed for avoidance (Fig. 6b–d, Supplementary Fig. 12). Neither FLP-3-1 nor FLP-3-4 were unable to rescue the avoidance phenotype on their own (Fig. 6b–d, Supplementary Fig. 12), supporting the flp-3(ok3625) data and further suggesting that these two peptides are insufficient to maintain wild-type behavior (Supplementary Fig. 4). This is surprising, as each peptide exhibits the highest affinities for FRPR-16, and two of the three highest affinities for NPR-10 (Fig. 4).

We also tested FLP-3-10 (STVDSSEPVIRDQ), which exhibits a lack of consensus sequence and an inability to activate either NPR-

10 or FRPR-16. The non-RFamide peptide was unable to rescue the avoidance phenotype (Fig. 6b, Supplementary Fig. 12a).

FLP-3-1 and FLP-3-4 differ in sequence only in their N-terminal amino acid (Figs. 3a, 4i). Similarly, a single amino acid change is all that distinguishes either peptide from FLP-3-2 (TPLGTMRFamide), which exhibits the second highest affinity for NPR-10 (Fig. 4b). We therefore tested FLP-3-2 for its ability to rescue the flp-3 lof phenotype. Surprisingly, this peptide was able to abolish the avoidance phenotype observed in flp-3 lof animals (Fig. 6b, Supplementary Fig. 12a), although it was not able to restore the animal's ability to be attracted to ascr#8 (Fig. 6 c, d, Supplementary Fig. 12c). The only difference being the presence of a threonine in that position of the peptide, we hypothesized that this may be the required component to suppress the avoidance behavior.

Peptide FLP-3-9 (NPENDTPFGTMRFamide) is a naturally occurring FLP-3 peptide that contains a threonine in the same location of the peptide. The N-terminus is capped with a NPEND sequence, and the lysine conserved in FLP-3-1, 3-2, and 3-4 is mutated to a phenylalanine. However, when flp-3 lof animals were fed NPENDTPFGTMRFamide, they not only displayed lack of avoidance to ascr#8, but also a full rescue of their ability to be attracted to the cue (Fig. 6, Supplementary Fig. 12).

To further validate the ability of this threonine (T) to suppress avoidance of ascr#8, we tested the ability of FLP-3-7 (SAEPFGTMR-Famide) to restore wild-type behavior (Fig. 6e–h). We observed that FLP-3-7 neither suppresses avoidance (Fig. 6h, Supplementary Fig. 13) nor drives attraction (Fig. 6f, g, Supplementary Fig. 13). However, when the ninth amino acid from the C-terminal, a glutamate (G), is replaced with a threonine, FLP-3-7T (Fig. 6e), we observed a suppression of the avoidance behavior (Fig. 6e, Supplementary Fig. 13), supporting our hypothesis that this threonine is critical for suppressing male-specific avoidance of ascr#8.

While FLP-3-9, which contains this threonine, can drive attraction, we observed that FLP-3-7T, which contains an SA sequence upstream of the threonine in place for FLP-3-9's NPEND sequence, is unable to drive attraction to ascr#8 (Fig. 6e–g, Supplementary Fig. 13). These results support our hypothesis that the NPEND sequence of FLP-3-9 is critical in regulating the attractive behavior.

Together, these data argue that the threonine in the ninth position from the C-terminus is critical for suppression of the basal avoidance response (Fig. 7a), as both FLP-3-2 and FLP-3-9, along with the modified FLP-3-7T peptide, were all capable of doing so, while FLP-3-1 and FLP-3-4 could not (Fig. 6, Supplementary Figs. 12, 13). Likewise, the NPEND sequence in FLP-3-9 may convey further specificity to the peptide, allowing it to drive attraction to the pheromone (Fig. 7a). It is not merely the presence of terminal amino acids that drives attraction, as the short sequence present in FLP-3-7T was unable to do so (Fig. 6).

**Discussion**
Our results reveal a complex mechanism regulating the sex-specific behavioral response to a pheromone guided through the interaction of at least two peptides encoded by a single-neuropeptide precursor gene and two divergent GPCRs. We show that two distinct NP/NPR modules driven by the single-neuropeptide gene flp-3 serve to drive sex-specific attraction to the male-attracting pheromone, ascr#8[14,28] (Fig. 2).

We developed and validated our novel single worm behavioral assay (SWAA), which confirmed that male C. elegans are indeed attracted to different pheromone cues (Fig. 1 and Supplementary Figs. 1, 2). Previous attraction assays, such as SRA (Supplementary Fig. 1), resulted in attraction values that were skewed due to male–male contact. The SWAA overcomes those caveats as it

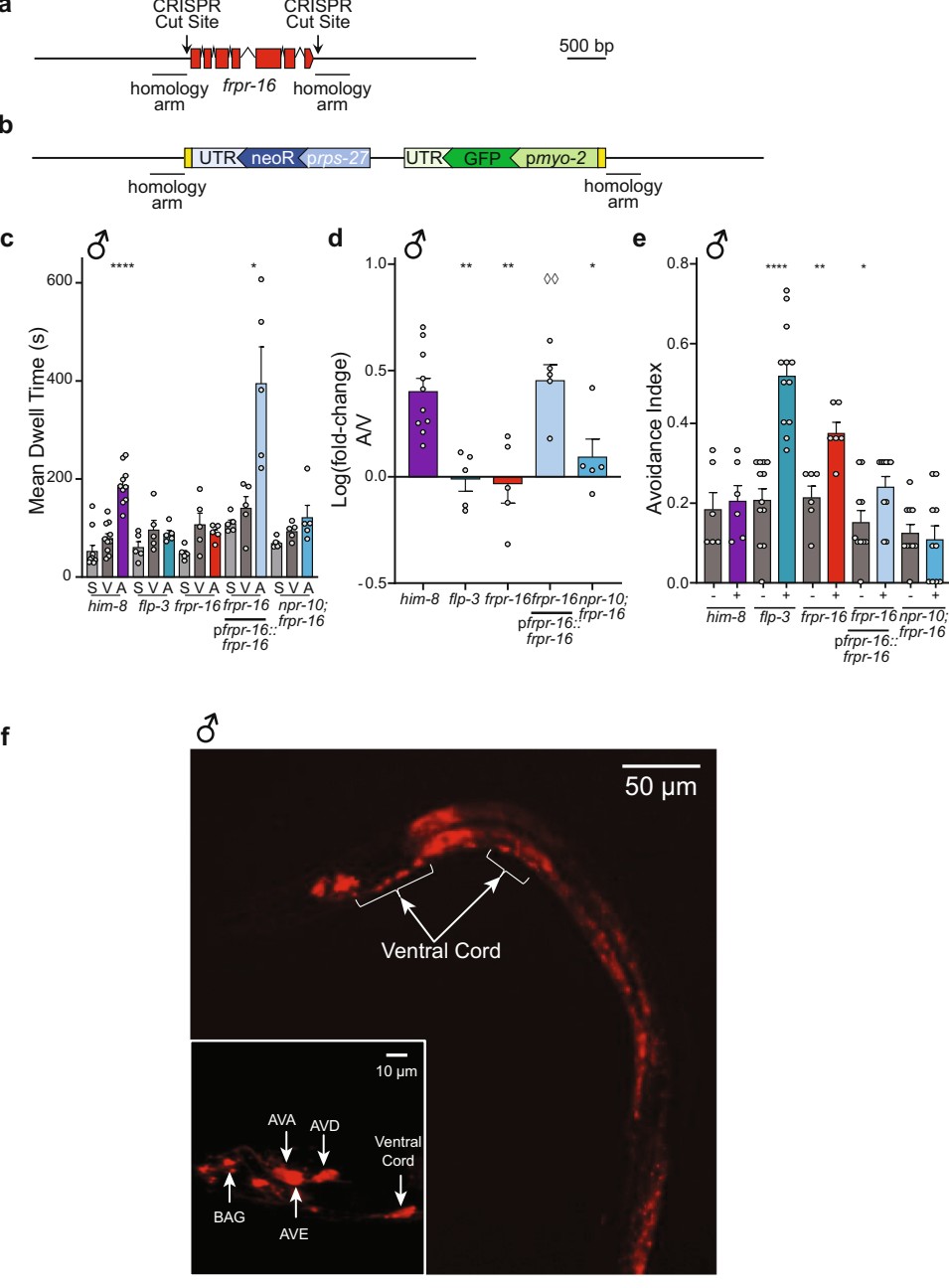

**Fig. 5 FRPR-16 is required for the male behavioral response to ascr#8. a**, **b** Design of CRISPR/Cas9-mediated *frpr-16* null mutation construct. **a** The wild-type gene, with CRISPR cut sites marked, along with 450 bp homology arm regions. **b** The mutant gene sequence, consisting of an inverted cassette driving loxP flanked p*myo-2*::GFP and p*rps-27*::neoR expression. **c**, **d** *frpr-16 lof* male animals display loss of attraction to ascr#8 and rescue of *frpr-16* under its endogenous promoter rescues the attraction phenotype. *frpr-16;npr-10* double mutant animals do not display an additive phenotype in terms of attraction **c** Raw dwell time and **d** log(fold-change) values. **e** *frpr-16 lof* male animals display increased avoidance to ascr#8 and rescue of frpr-16 restores avoidance to wild-type male animals. *frpr-16;npr-10* double mutant animals do not display avoidance to ascr#8. **f** Expression pattern of p*frpr-16*::*frpr-16*::SL2::mCherry in male *C. elegans* at ×20 magnification. Localization within the ventral cord denoted. **f** (inset), Amphid localization of p*frpr-16*::*frpr-16*::SL2::mCherry at ~×120 within the reverse locomotion command interneurons, AVA, AVE, and AVD, as well as the BAG neuron (anterior to the nerve ring). In panel **d** ◊◊ $p < 0.01$ for *frpr-16 lof* mutant versus transgenic rescue. ♂ denotes male data.

measures the attractive properties of individual animals in each spot (Fig. 1, Supplementary Fig. 2). It also helps calculate the percentage attractiveness of the cue adding another parameter to measure the robustness of the cue. Our SWAA suggests that both *him-5* and *him-8* males are equally attracted to ascr#8 (Fig. 1, Supplementary Fig. 2). This addresses one of the caveats of the SRA results, wherein *him-5* and *him-8* males responded differentially to ascr#8, with *him-8* males being significantly more

attracted than their *him-5* counterparts (Supplementary Fig. 1). This may be due to a high degree of male–male contact within the ascr#8 spot, but not the vehicle spot in SRA (Supplementary Fig. 1). We additionally confirmed that another previously described male attractant ascaroside, acsr#3, maintains its ability to attract males in the SWAA (Supplementary Fig. 5). We also demonstrate that hermaphrodites, which have previously been shown to leave food rarely in comparison to males[60], overcome

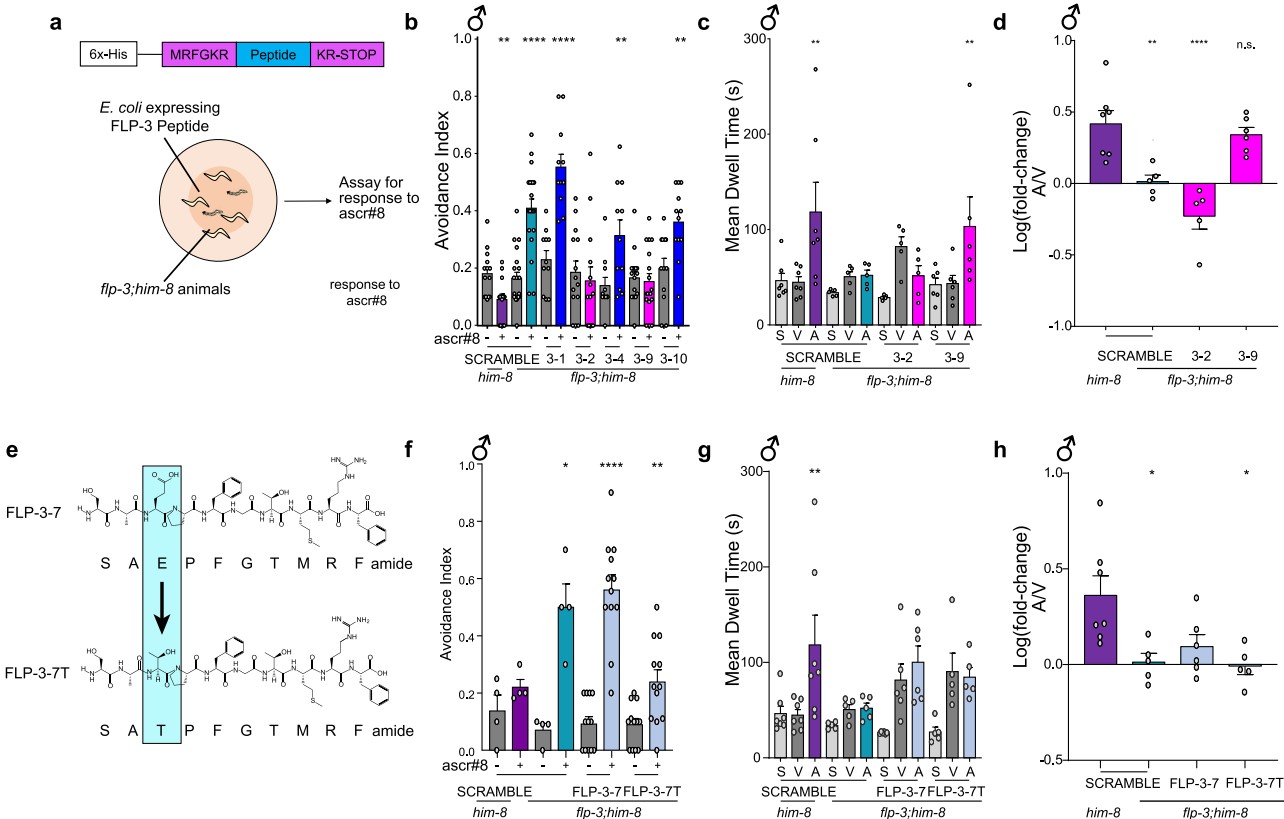

**Fig. 6 Peptide feeding rescues wild-type behavior and reveals two active peptides within the FLP-3 precursor. a** Overview of rescue-by-feeding paradigm: (Top) A plasmid in generated which encodes the peptide of interest is flanked by EGL-3 cleavage sites, with a 6x-His tag upstream. (Bottom) *flp-3 lof* animals are raised on bacteria expressing the FLP-3 peptide of interest for 72-96 h and are assayed as young adults. **b** Avoidance behavior of *him-8* and *flp-3* animals raised on scramble peptide or specified FLP-3 peptides. *him-8* and *flp-3* males fed SCRAMBLE display their characteristic avoidance to ascr#8. *flp-3* males fed peptides FLP-3-2 or FLP-3-9 restore avoidance to wild-type levels. **c, d** *flp-3* male animals fed FLP-3-9 peptides rescues attraction behavior to ascr#8 **c** Raw dwell time and **d** log(fold-change) values for *him-8* and *flp-3* males fed peptides FLP-3-2 or FLP-3-9. **e** Mutational schematic of FLP-3-7 to generate FLP-3-7T: the glutamate ("E") in position 9 was mutated to a threonine ("T"). **f** *flp-3 lof* male animals raised on FLP-3-7T suppress avoidance to ascr#8 suggesting the importance of amino acid threonine in mediating avoidance. **g, h** *flp-3* animals raised on FLP-3-7 and FLP-3-7T display no rescue of attractive behavior to ascr#8. **g** Raw dwell time and **h** log(fold-change) of *flp-3* animals raised on FLP-3-7 and FLP-3-7T. ♂ denotes male data.

spatial control effects reminiscent of 'edge effects'[61]—wherein animals spend more time along the edge rather than in the throughout the well—and instead spend more time the vehicle control more often (Fig. 1d). We observe that this dwell time is dramatically reduced in the presence of the pheromone, supporting the findings that hermaphrodites avoid ascr#8 (Fig. 2f).

Peptidergic modulation of neural circuits as long been hypothesized as complex[62], and here we elucidate the recruitment of two neuropeptide/neuropeptide-receptor (NP/NPR) modules, FLP-3/NPR-10 and FLP-3/FRPR-16, that both serve to regulate the nervous system in sensing and respond to asr#8 in a sex-specific manner. Our results elucidate that not all peptides encoded by the FLP-3 propeptide are involved in the regulation of the sex-specific circuit, but rather a subset function through two unique NP/NPR modules to drive the behavioral response. This suggests that the complexity of the expansive class of FMRFamide-related peptide (FLP) genes in *C. elegans*, of which there are 31 genes encoding over 70 unique peptides[42], function through an even greater number of NP/NPR modules to drive specific behavioral or physiological states. FLPs have been identified as regulators of a variety of behavioral and sensory mechanisms, including locomotion[19], egg-laying[42], gas sensing[63], sleep[29], and mating[22,64]. Here, we show that *flp-3* functions to coordinate ascr#8 sensation with attractive behavior.

Previous studies have linked the entirety of the gene to a receptor based on binding studies and full transgenic rescue[29,65,66]. Here, we employ a rescue-by-feeding assay, following the design of RNAi feeding protocols, to rescue individual peptides[37]. While "feeding" of peptides through soaking is a valid approach, there are many constraints on such approaches, the most prominent being the ability to acquire purified peptides[29,65]. Using our rescue-by-feeding approach, we provide access to the peptide to the worms directly through their food source. This approach enables new avenues for characterizing the roles of the other neuropeptides in mediating diverse cellular and organismal processes.

Combining biochemical receptor activation studies with behavioral rescue-by-feeding assays, we have been successful in elucidating discrete neuropeptide signaling modalities within the complex FLP-3 signaling system. The involvement of two evolutionarily divergent receptors in sensing specific FLP-3 neuropeptides suggests that the sex-specific behavioral response to ascr#8 module is a result of the activity of two distinct NP/NPR modules that mediate both attractive and repulsive properties of the small molecule. Two GPCRs respond to FLP-3 peptides to function in the behavioral response to ascr#8: the previously identified NPR-10, and the novel FRPR-16 (Figs. 3–5). Both exhibit high potencies for multiple FLP-3 peptides, although our single-peptide rescues have shown that FLP-3-2 and FLP-3-9 are required for the wild-type response to ascr#8 (Fig. 6), while FLP-

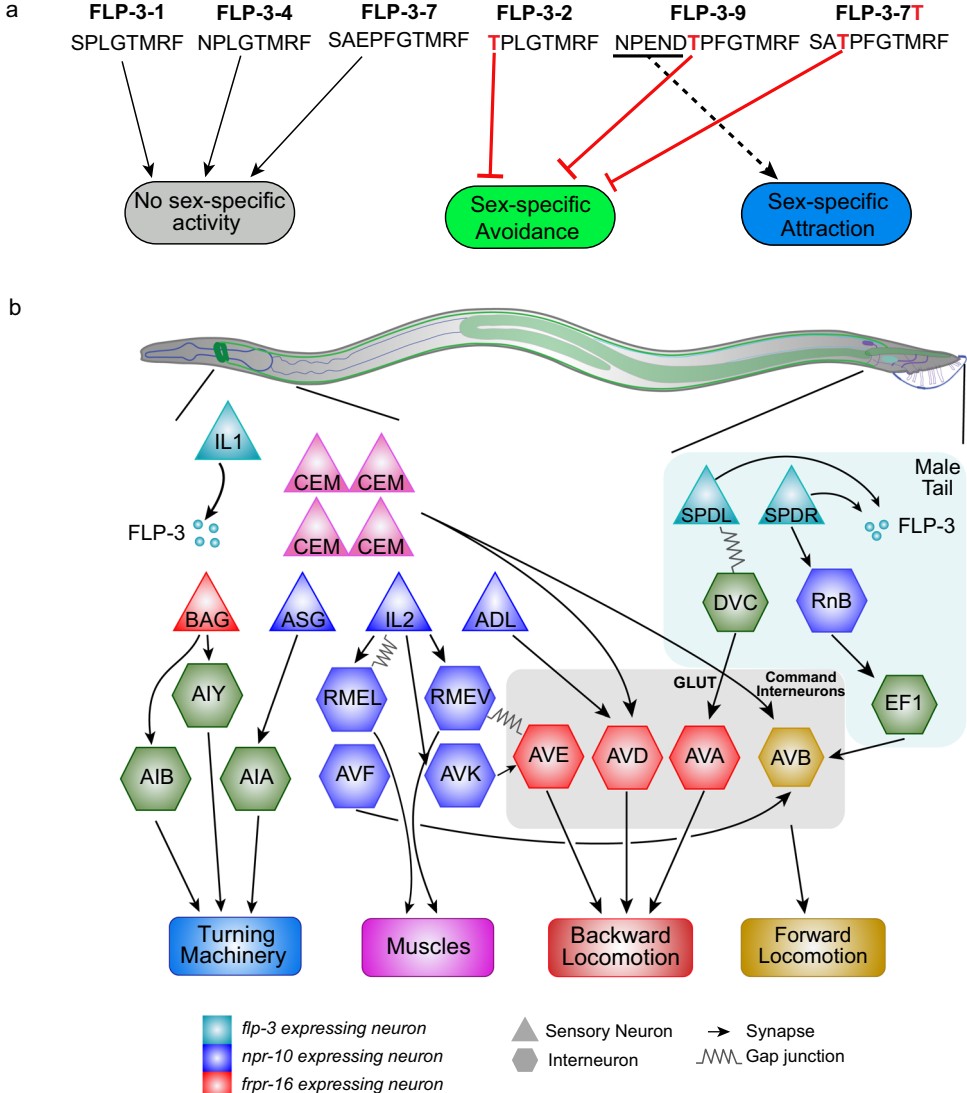

**Fig. 7 Two FLP-3 NP/NPR modules mediate the male behavioral response to ascr#8. a** Sequence specificity of FLP-3 peptide function. FLP-3-1, FLP-3-4, and FLP-3-7 exhibit no sex-specific effects on ascr#8 behavioral response. The threonine in the 9th position of FLP-3-2 and FLP-3-9, and in the mutated FLP-3-7T (red) is required for suppressing the sex-specific avoidance of ascr#8. The NPEND sequence of FLP-3-9 drives male attraction to ascr#8. **b** Schematic of the neural circuit regulating male responses to ascr#8 pheromone. ascr#8 is sensed by the male-specific CEM neurons. The neuropeptide gene *flp-3* is expressed in the IL1 neurons in the head and the SPD spicule neurons in the tail (teal). Following release of processed FLP-3 peptides, FLP-3-2 and FLP-3-9 are sensed by NPR-10 and FRPR-16 expressing neurons (blue and red, respectively) to mediate dwelling in the mating pheromone by modulating forward and reverse locomotion. Green neurons are connections inferred from the male synaptic connectome[73]. Gold denotes the command interneuron AVB and the forward locomotory circuitry.

3-1, FLP-3-4, and FLP-3-10 are not. As such, multiple NP/NPR modules are implicated in the ascr#8 behavioral response. Further studies will allow for further separation of FLP-3-2 and FLP-3-9, and how they interact with NPR-10 and FRPR-16.

Interestingly, these two high-potency FLP-3 receptors are extremely divergent in their evolutionary history. NPR-10 is most related to other "NPR" *C. elegans* receptors that evolved from the same family as the *Drosophila melanogaster* Neuropeptide F Receptor family[55,67], and exhibits predicted NPY activity. FRPR-16, however, is more closely related to the fly FMRFamide Receptor[67]. Interestingly, while some *C. elegans* FRPR receptors function as FLP receptors[19,64,65], at least one receptor within the same evolutionary clade (DAF-37) acts as a chemosensor for pheromones[68]. The evolutionary distance between NPR-10 and FRPR-16 suggests that these two receptors have undergone convergent evolution.

The presence of FRPR-16 in the amphid premotor interneurons responsible for backwards locomotion (Fig. 5) suggests the FLP-3/FRPR-16 module serves to mediate reversals during ascr#8 sensation (Fig. 7b). Conversely, while FRPR-16 is confined to a small, yet biologically specific subset of neurons in the head of the animals, NPR-10 exhibits more promiscuous expression that innervates both forward and reverse locomotion circuitries (Fig. 3e, f). As such, while the FLP-3/FRPR-16 module specifically modulates reversals, the FLP-3/NPR-10 module may instead serve to balance both forward and backwards locomotion in response to ascr#8, allowing the animal to interrogate its surroundings more thoroughly.

We argue that the loss of both of these modules that underlies the behavioral response in *npr-10;frpr-16* double mutant animals to ascr#8. The loss of only one module results in a skewed behavioral response, observed as aversion to ascr#8 (Figs. 3, 5).

This aversion likely arises from the absence of amphid NPR-10 leading to aberrant regulation of turning machinery (through ASG-AIA connections) and muscle innervation (via RME neurons). In the tail, lack of NPR-10 seems to prevent EF1 activation, and thereby allowing forward locomotion to continue. Meanwhile, loss of only FRPR-16 will lead to the inability of FLP-3 to regulate backward locomotion machinery (Fig. 7b). However, the loss of both the FLP-3/NPR-10 and FLP-3/FRPR-16 modules abolishes the ability of the animal to response to ascr#8 at all, observed as neither attraction nor aversion (Fig. 5). We post that the absence of both receptors results in the inability of FLP-3 to suppress forward locomotion (through NPR-10) nor drive backwards locomotion (via FRPR-16) in response to acsr#8. Future studies incorporating cell-specific rescue of both NPR-10 and FRPR-16 will further elucidate this circuitry.

While we only investigated the activity of single peptides in this study, FLP-3 is a complex gene encoding 10 discrete peptides. Not all peptides are sufficient to rescue the ascr#8 behavior on their own, they may instead serve a synergistic role to "active" peptides. Our rescue-by-feeding approach makes it easy to perform combinatorial studies of peptides[59], allowing for the elucidation of such synergistic peptide function. However, other reasons for the lack of rescue are possible, including rapid rates of degradation of the rescue peptide RNA. Future studies should make note to carefully dissect negative results, using either peptide soaking[29,65], or comparison to in vivo expression with partial null mutants. We pursued the second course of action, discerning that FLP-3-1 and FLP-3-4 are insufficient to suppress ascr#8 avoidance in either the flp-3(ok3625) allele (Supplementary Fig. 4), or in our rescue-by-feeding paradigm (Fig. 6).

Our findings highlight the complexity of neuromodulators regulating behavioral states in both invertebrates and vertebrates. Multiple hormonal receptors are involved in regulating the induction of A. aegyptii ecdysteroid hormone production through two different neuropeptide signaling systems: ILP3 initiates digestion of the blood meal[69], while OEH stimulates oocyte yolk uptake[70]. Melanin-concentrating hormone is a neuropeptidergic hormone that promotes appetite and feeding behaviors in mice in a sex-dependent manner[71]. Meanwhile, age-dependent changes in levels of Neuropeptide F result in the promotion of survival-benefiting appetitive memory in Drosophila, concurrent with the impairment of memories associated with insufficient survival benefits[72].

Sex-specific behaviors arise from processing specialized olfactory cues emitted by their conspecifics[2]. Odor processing within each sex is mediated by flexible neural circuits and neuromodulation enables neural networks to adapt behaviors under fluctuating external and internal environmental states. So how is this adaptability achieved? Our studies demonstrate that specific peptides encoded by a single-neuropeptide gene, activate evolutionarily divergent receptors resulting in fine-tuned sex-specific behavioral responses to small-molecule pheromones. These NP/NPR modules are expressed within specific neurons of the nervous system of the two sexes, mediating overlapping behavioral outputs, by simultaneously suppressing an avoidance response and driving an attractive response (Fig. 7b). The CEM neurons, which act as a primary site of ascr#8 sensation, synapse onto both forward (AVB) and backward (AVD) locomotory neurons, the activity of which are modulated by these NP/NPR modules both indirectly and directly, respectively (Fig. 7b)[73]. These findings highlight the complexity of peptidergic modulation of the nervous system, wherein individual peptides either from a single gene or multiple genes modulate opposing behaviors, through multiple NP/NPR modules. Using our rescue-by-feeding paradigm[59], we can unravel the function of discrete peptides, enhancing our understanding of pathways of extra synaptic information flow in the complex functional connectome.

## Methods

**Strains.** Strains were obtained from the *Caenorhabditis* Genetics Center (University of Minnesota, MN), the National BioResource Project (Tokyo Women's Medical University, Tokyo, Jagan), Chris Li at City University of New York, Paul Sternberg at the California Institute of Technology, Ding Xue at University of Colorado Boulder, and Maureen Barr at Rutgers University. The novel allele of *frpr-16* was generated via CRISPR editing using previously discussed methods[58]. Strains were crossed with either *him-5* or *him-8* worms to generate stable males prior to testing. See Supplementary Table 1 for a comprehensive list of strains used in this study.

### Vector generation

*Peptide constructs.* DNA oligos containing the sequence for the peptides of interest were generated using Integrated DNA Technologies' Ultramer synthesis service. The DNA sequence encoding the peptide sequence was flanked with sequences encoding EGL-3 cut sites (MRFGKR upstream, and KRK-STOP) downstream[59]. These sites were then flanked with Gateway Cloning sites attB1 and attB2. Annealed oligos were then used to perform a BP reaction with pDONR p1-p2 to generate the pENTRY clones. These vectors where then recombined with pDEST-527 (a gift from Dominic Esposito (Addgene plasmid # 11518) in LR reactions to generate the expression clones[59]. The SCRAMBLE control was generated in an identical manner, with the sequence between the cut sites being amplified from pL4440 (provided by Victor Ambros, University of Massachusetts Medical School, MA).

*Fusion constructs.* DNA for the *flp-3, npr-10,* and *frpr-16* promoter and coding regions were isolated from *C. elegans* genomic DNA via PCR.

In generating the *flp-3* rescue product, PstI and BamHI restriction sites added onto the isolated fragments were introduced through primer design. PCR amplicons and the Fire GFP Vector, pPD95.75 (kindly provided by Josh Hawk, Yale University, CT), were digested with PstI and BamHI enzymes. Products were ligated together to generate JSR#DKR18 (p*flp-3*::*flp-3*::GFP). The *flp-3* expression analysis construct, JSR#DKR34 (p*flp-3*::*flp-3*::SL2::mCherry) was generated by Genewiz. The promoter-gene fragment of *npr-10* was generated by Gibson Assembly to GFP (from pPD95.75) and as a linear fusion. The rescue-fusion construct of *frpr-16* was achieved by fusing the promoter and gene sequence to mCherry isolated from JSR#DKR34 via Gibson Assembly.

See Supplementary Table 2 for a complete plasmid list, and Supplementary Table 3 for primer and Ultramer sequences.

**Transgenic animals.** CB1489 animals were injected with JSR#DKR18 (p*flp-3*::*flp-3*::GFP at 20 ng/µL), using p*unc-122*::RFP (at 20 ng/µL) (kindly provided by Sreekanth Chalasani at the Salk Institute, CA) as a co-injection marker to generate JSR81 (*him-8*(e1489);worEx17[p*flp-3*::*flp-3*::GFP; p*unc-122*::RFP]). JSR81 was then crossed with JSR99 to generate JSR109 (*flp-3*(pk361);*him-8*(e1489);worEx17[p*flp-3*::*flp-3*::GFP; p*unc-122*::RFP]).

PS2218 animals were injected with JSR#DKR34 (p*flp-3*::*flp-3*::SL2::mCherry at 25 ng/µL), using p*unc-122*::GFP (at 50 ng/µL) as a co-injection marker to generate JSR119 (*dpy-20*(e1362);*him-5*(e1490);syls33[HS.C3(50 ng/µL) + pMH86(11 ng/µL)];worEx21[p*flp-3*::*flp-3*::SL2::mCherry; p*unc-122*::GFP]).

JSR102 animals were injected with a linear fusion product (p*npr-10*::*npr-10*::GFP at 25 ng/µL), alongside p*unc-122*::RFP (at 50 ng/µL) as a co-injection marker to generate JSR126 (*npr-10*(tm8982);*him-8*(e1489);worEx37[p*npr-10*::*npr-10*::GFP, p*unc-122*::RFP]). This was crossed with PT2727 (myIS20 [p*klp-6*::tdTomato + pBX], a gift from Maureen Barr) to generate JSR138 (myIS20 [p*klp-6*::tdTomato, pBX]; *npr-10*(tm8982);*him-8*(e1489);worEx37[p*npr-10*::*npr-10*::GFP, p*unc-122*::RFP]).

JSR103 animals were injected with a linear fusion product (p*frpr-16*::*frpr-16*::SL2::mCherry at 25 ng/µL), alongside p*unc-122*::GFP (at 50 ng/µL) as a co-injection marker to generate JSR133 (*frpr-16*(gk5305[loxP + p*myo-2*::GFP::*unc-54* 3′ UTR + p*rps-27*::neoR::*unc-54* 3′ UTR + loxP]);*him-8*(e1489);worEx41[p*frpr-16*::*frpr-16*::SL2::mCherry, p*unc-122*::GFP]).

Injections for JSR81 were generously performed by the Alkema Lab at UMass Medical School. Injections for JSR119, JSR126, and JSR133 were performed by In Vivo Biosystems (formerly NemaMetrix).

**Chemical compounds.** The ascarosides ascr#3 and ascr#8 were synthesized as described previously[28,33]. Peptides used in in vitro GPCR activation assays were synthesized by GL Biochem Ltd.

**Spot retention assay.** Assays were performed as described previously[14,33]. Fifty to sixty larval-stage 4 (L4) males were segregated by sex and stored at 20 °C for 5 h to overnight to be assayed as young adults. For hermaphrodite trials, young adult hermaphrodites were segregated 1.5 h prior to testing. 0.6 µL of vehicle control or ascaroside #8 was placed in each scoring region (Supplementary Fig. 1a). As the working stock of ascaroside #8 was made in MilliQ-purified ultrapure H$_2$O, this was used as the vehicle control. Five animals were placed on each "X" the assay plate (Supplementary Fig. 1a), which was then transferred to a microscope

containing a camera and recorded for 20 m. Each strain and sex were assayed over five plates per day on at least three different days.

**Single worm assay**. The outer forty wells of a 48-well suspension culture plate (Olympus Plastics, Cat #: 25-103) were seeded with 200 μL of standard NGM agar. To prepare the plates for the assay, they were acclimated to room temperature, at which point each well was seeded with 65 μL of OP50 *E. coli*. The assay plates were then transferred to a 37 °C incubator with the lid tilted for 4 h to allow the bacterial culture to dry on the agar. Once the bacterial culture dried, the lid was replaced the plate was stored at 20 °C until used in the assay. Fifty to sixty L4 worms were segregated by sex and stored at 20 °C for 5 h to overnight to be assayed as young adults. 0.8 μL of either vehicle control or ascaroside #8 was placed in the center of the well corresponding to that condition within the quadrant being tested, following a random block design (Fig. 1a). A single worm was placed in each of the 10 wells to be assayed, and the plate was transferred to a light source and camera and recorded for 15 m. This process was repeated for all four quadrants. Each strain and sex were assayed over five plates assayed on at least three different days.

**Raw dwell time**. Raw dwell time values were calculated by subtracting the time a worm exited the cue (center of the well in spatial controls), from the time it entered, as in the SRA[14]. This was determined per visit, and the average dwell time was calculated for each animal in the quadrant. Averages of the four-quadrant means were determined per plate, and a minimum of five plates were assayed per strain/condition. The mean raw dwell time across five plates was calculated and used for statistical analyses and graphical display.

**Log(fold-change)**. The average dwell time in the ascaroside was divided by the average dwell time within the vehicle control per plate to generate a fold-change. To transform the data, the log of this fold-change was taken, and the average log(fold-change) was used for statistical analyses and graphical display.

**Visit count**. The number of visits per-worm was calculated, and the average visit count determined per quadrant, and per plate. The average visit count across five plates was calculated and used for statistical analyses and graphical display.

**Percent attractive visits**. An "attractive visit" was first determined for each plate as any visit greater than two standard deviations above the mean dwell time within the vehicle control for that plate. Any individual visit meeting this threshold was scored as a "1", and any below was scored a "0". The percent visits per-worm that were attractive was determined, and the average of each quadrant taken. The four-quadrant values were then averaged to generate plate averages. The average percent of attractive visits across five plates was calculated and used for statistical analyses and graphical display.

**Avoidance assay**. Assays were performed as described previously[4,11,41]. Fifty to sixty L4 worms were segregated by sex and stored at 20 °C for 5 h to overnight to be assayed as young adults. One to four hours prior to the assay, the lids of unseeded plates were tilted to allow any excess moisture to evaporate off the plates. At the time of the assay, 10 or more animals were transferred onto each of the dried, unseeded plates. A drop of either water or 1 μM ascr#8 was placed on the tail of forward moving animals, and their response was scored as either an avoidance response, or no response. The total number of avoidances was divided by the total number of drops to generate an avoidance index for that plate. This was repeated for at least 10 plates over at least three different days.

### Statistical analyses

*Spot retention assay*. Statistical comparisons within each strain were made by Paired *t*-tests. For comparisons between strains/conditions, the data was transformed as described previously[38]. In short, the data were transformed to have only non-zero data for the calculation of fold-changes. This was done using a Base 2 Exponentiation ($2^n$, where n is equal to the dwell time). The log (base 2) of the fold-changes of these transformed values was used to allow for direct comparisons between strains of the same background (i.e., *him-5* and *osm-3;him-5*) using a Student's *t*-test. *p* values are defined in respective figure captions, with thresholds set as: *$p < 0.05$, **$p < 0.01$, ***$p < 0.001$, ****$p < 0.0001$.

*Single worm assay*. Statistical comparisons within each strain/sex (spatial, vehicle, ascaroside) were made by Repeated Measured ANOVA with the significance level set at 0.05, followed by multiple comparisons using Bonferroni correction. For comparisons between strains/sexes, the spatial control dwell times were compared using a one-way ANOVA followed by a Dunnett's correction to confirm that mutations of interest had no effect on the amount of time animals naturally spent in the center of the well. To directly compare strains, a fold-change was calculated by dividing the ascaroside by vehicle dwell times for each assay. This was then

transformed by taking the log (base 10) of the fold-change. Comparisons were then made by one-way ANOVA followed by multiple comparisons using Dunnett's correction. *p* values are defined in respective figure captions, with thresholds set as: *$p < 0.05$, **$p < 0.01$, ***$p < 0.001$, ****$p < 0.0001$.

*Avoidance assay*. Statistical comparisons within each strain were made by paired *t*-test against a significance level set at 0.05. For comparisons between strains/conditions, comparisons were made by One-Way ANOVA, followed by multiple comparisons using Bonferroni correction. *p* values are defined in respective figure captions, with thresholds set as: *$p < 0.05$, **$p < 0.01$, ***$p < 0.001$, ****$p < 0.0001$.

*In vitro GPCR activation assay*. The GPCR activation assay was performed as previously described[66,74]. Briefly, *npr-10* and *frpr-16* cDNAs were cloned into the pcDNA3.1 TOPO expression vector (Thermo Fisher Scientific). A CHO-K1 cell line (PerkinElmer, ES-000-A24) stably expressing apo-aequorin targeted to the mitochondria (mtAEQ) and human Gα16 was transiently transfected with the receptor cDNA construct or the empty pcDNA3.1 vector using Lipofectamine LTX and Plus reagent (Thermo Fisher Scientific). Cells were shifted to 28 °C one day later and allowed to incubate for 24 h. On the day of the assay, cells were collected in BSA medium (DMEM/Ham's F12 with 15 mM HEPES, without phenol red, 0.1% BSA) and loaded with 5 mM coelenterazine h (Thermo Fisher Scientific) for 4 h at room temperature. The incubated cells were then added to synthetic peptides dissolved in DMEM/BSA, and luminescence was measured for 30 s at 496 nm using a Mithras LB940 (Berthold Technologies) or MicroBeta LumiJet luminometer (PerkinElmer). After 30 s of readout, 0.1% triton X-100 was added to lyse the cells, resulting in a maximal calcium response that was measured for 10 s. After initial screening, concentration-response curves were constructed for HPLC-purified FLP-3 peptides by subjecting the transfected cells to each peptide in a concentration range from 1 pM to 10 μM. Cells transfected with an empty vector were used as a negative control. Assays were performed in triplicate on at least two independent days. Concentration-response curves were fitted using Prism v. 7 (nonlinear regression analysis with a sigmoidal concentration-response equation).

**Generation of a Null *frpr-16* mutant by CRISPR mutagenesis**. The *frpr-16* CRISPR/Cas9 knockout was provided by the Vancouver node of the International *C. elegans* Consortium. The mutation was generated following previously described techniques[58]. In short, a 1685 bp region containing the coding sequence, as well 52 bp upstream and 60 bp downstream, was removed from the genome, and replaced with a trackable cassette containing p*myo-2*::GFP and a neomycin resistance gene (Fig. 5a, b). The flanking sequences of the mutated sequence are TCA TAATTGTTTGTTTGACAAAAACCGGGA and GGTGGAAACGGAAATGAAA GAAAAAACCGA. PCR confirmation of gene replacement with cassette was performed via four sets of PCR reactions checking the upstream insertion site and the downstream insertion site in the mutant strain, and a test for wild-type sequence in both mutant and wild-type strains. A band is present on the gel in wild-type samples, with no band present in the mutant, as a primer sequence is removed with the cassette insertion.

See Supplementary Table 3 for primer sequences.

**Peptide rescue**. SCRAMBLE control or FLP-3 peptide constructs were grown overnight in LB media containing 50 μg/μL ampicillin at 37 °C and diluted to an OD$_{600}$ of 1.0 prior to seeding on NGM plates containing 50 μg/μL ampicillin and 1 mM IPTG. The 75 μL lawn was left to dry and grow overnight at room temperature before three L4 animals were placed on the plates. Males were selected for testing in the same manner as described above but were isolated onto plates also seeded with the same peptide on which they had been reared[59]. Animals were then assayed using either the avoidance assay or single worm assay.

**Imaging**. Animals were mounted on a 2% agar pad and paralyzed using 1 M sodium azide on a microscope slide, as described previously[11].

Images for amphid *flp-3* expression were acquired using a Zeiss LSM700 confocal microscope. Final images were obtained using a ×63 oil objective with a ×1.4 digital zoom, for a final magnification of ~×90. Tail images were acquired using a Zeiss Apotome at ×40 oil objective.

Images of *npr-10* and *frpr-16* expression were acquired using a Zeiss LSM510 Meta inverted confocal microscope. Final images of *npr-10* were obtained using a ×63 oil objective with a ×0.8 digital zoom, for a final magnification of ~×50. Final images of *frpr-16* were obtained at either ×20 (air objective) or ×63 (oil) with a 2× digital zoom, for a final magnification of ~×125.

**Reporting summary**. Further information on research design is available in the Nature Research Reporting Summary linked to this article.

## Data availability
The authors declare that the data supporting the findings of this study are available within the paper and its supplementary information files.

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

## Acknowledgements
We thank the *Caenorhabditis* Genetics Center, which is funded by the NIH Office of Research Infrastructure Programs (P40 OD01044), as well as Dr. Paul Sternberg (Cal-Tech), Dr. Chris Li (CUNY), the National BioResource Project, Dr. Ding Xue (UC Boulder), and Dr. Maureen Barr (Rutgers) for providing strains. The synthetic ascr#8 utilized in this study was generously provided by Frank Schroeder (Cornell University). We want to thank Dr. Don Moerman (UBC; C. elegans Knockout Facility) for providing the *frpr-16* knockout. We thank InVivo Biosciences for generating transgenic animals by injection. We thank Kate Pearce and Suzanne Scarlatta for assistance in confocal fluorescent imaging the *npr-10* and *frpr-16* reporter strains, as well as Dr. Jeremy Florman of Mark Alkema's lab for assistance in *flp-3* reporter imaging. We also would like to thank Dr. Victor Ambros (UMass Medical School), Dr. Shreekanth Chalasani (Salk Institute), Dr. Josh Hawk (Yale), and Dr. Dominic Esposito (Addgene) for providing us with the necessary plasmid reagents. We acknowledge and thank Profs. Frank Schroeder and Shreekanth Chalasani, as well as members of the Srinivasan lab for their feedback on this manuscript. Funding for this work was provided under the Research foundation Flanders (FWO) project grant G0C0618N (I.B.), and the National Institutes of Health grants R01 NS107475 (M.J.A.), and R01 DC016058 (J.S.).

## Author contributions
D.K.R. performed behavioral assays, contributed to the single worm assay design, performed all molecular biology, performed and designed peptide rescue experiments, performed the statistical analyses, and led manuscript writing and revision. E.J.M., H.T.N., and A.N.R. contributed to the behavioral assays and as well as manuscript revisions. A.N.R. and C.S.M. generated the materials and performed experiments related to FLP-3-7. E.V. and I.B. performed the GPCR activation assays, as well as contributed to manuscript revisions. I.B. also provided funding for the GPCR activation studies. W.J. contributed to injections of the *flp-3* rescue, while M.J.A. provided comments on manuscript. R.J.G. assisted in the development of the single worm assay design. J.S. and D.K.R. wrote the manuscript with input from M.A. and R.J.G.

## Competing interests
The authors declare no competing interests.
