## [Peer Review File · Communications Biology]

Reviewers' comments:

Reviewer #1 (Remarks to the Author):

This is a very interesting paper by Reilly et al. where they find that several peptides encoded by the flp-3 gene determine the valence of the pheromone component ascr#8 in *C. elegans* males. One major strength of the work is the breadth of approaches they employ. They use a combination of new behavioural assays, genetics, gene expression analysis, genome editing, biochemistry and rescue-by-feeding to show that the peptides flp-3.2 and 3.9 act through their receptors npr-10 and frpr-16 to switch the behaviour of males towards ascr#8 from repulsion to attraction. In addition, they authors are able to dissect which peptides and perhaps even aminoacids are important for a two-step process of suppression of avoidance and promotion of attraction. How sensory perception and valence of stimuli are encoded in the nervous system at the molecular and cellular level is a major question in neurobiology and one that this journal's broad readership will be interested in.

Major coments:

- The authors refer to supplemental figure 2B to state that only a small percentage of animals (30-45%) exhibit attractive visits to the cue but the graph shows % of attractive visits per worm not % of worms visiting the spot. Where is the data showing % animals?

And also, what is the % of animals showing avoidance? This data seems important because the authors state that the CEMs are the only sensors of ascr#8 (for both attraction and avoidance) and that the 35-40% responses matches well the rate of CEM calcium transients upon ascaroside exposure.

- The authors conclude that the CEM neurons are the only source of ascr#8 chemosensation based on the fact that *ceh-30* mutants (which lack CEMs) are neither attracted to nor repelled by ascr#8. This conclusion is an overstatement. The experiment shows that the CEMs are indeed required for sensation, however, it doesn't exclude the possibility that other neurons may also be required (despite not being sufficient to drive behavioural responses in the absence of the CEMs). Furthermore, because npr-10, one of the receptors for flp-3 which is important for the behavioural responses to ascr#8, is expressed in ADL, a neuron shown to sense other ascarosides in a sexually dimorphic manner, the authors should test the requirement of ADL in ascr#8 sensation.

- Through the rescue-by-feeding experiments, the authors very nicely show that some peptides specifically drive attraction and others suppress avoidance. Based on the sequence of such peptides, they propose that a Threonin in the 9th position from the C terminus of the peptide is critical for suppression. This conclusion could be further strengthened with a direct manipulation: by testing whether switching the E in that position by a T in flp-3.7 and feeding it, rescues the avoidance phenotype of flp-3 mutants.

- Lines 301-302, fig. 5E – the npr-10 and frpr-16 double mutant has a somewhat surprising phenotype where the avoidance phenotype of single mutants is suppressed. The authors only state that the phenotype is not additive and therefore the receptors are not redundant. However, here or in the discussion, they should provide some potential explanations for the wildtype-like phenotype of the double mutant.

Minor comments:

- When they describe the expression pattern of flp-3 for the first time (p11 and 12) it is not clear whether IL1 and SPD are the only sites of expression and whether this is male-specific expression or whether hermaphrodites also have expression in IL1.

- The figures need to be better labelled stating whether the data refers to males or hermaphrodites; to ascr-8 or ascr-3; the expression patterns are of the ligand or the receptors; etc. Otherwise it gets a bit confusing.

- Line 212 – PQR is a sensory neuron, not an interneuron

Reviewer #2 (Remarks to the Author):

Males and hermaphrodites have differential responses to several pheromones in the nematode *C. elegans*. Males have an attractive response to the pheromone ascaroside #8 (ascr#8), while hermaphrodites show avoidance. Neuropeptides, such as FLP-3, 6, 12, and 19, are expressed in several male, sex-specific neurons. The authors premise that some pheromone responses are mediated through neuropeptide signaling. Response to ascr#8 was examined in this paper.

Animals were tested for their dwell time on and attraction to ascr#8. Wild-type males dwell on the ascr#8 spot for significantly more time than hermaphrodites, which avoid ascr#8. flp-6, 12, and 19 male mutants showed similar dwell times as wild-type males. By contrast, flp-3 male mutants showed significantly lower dwell times on and avoided ascr#8, suggesting that FLP-3 peptides act to suppress ascr#8 avoidance. The response of flp-3 mutants to ascr#3, a different chemoattractive pheromone, was unaffected, nicely showing that responses to ascr#8 and #3 are mediated by different pathways.

CEH-30 transcription factor is necessary for specification of the male-specific CEM neurons. Knockout (KO) of the CEMs in males did not result in an avoidance response to ascr#8 in a wild-type or flp-3 mutant background, suggesting that FLP-3 peptides are released from CEMs to suppress the avoidance response.

Several FLP-3 receptors have been previously identified: NPR-4, NPR-5, and NPR-10; the authors also classify the *Drosophila* homologue FRPR-16 as a FLP-3 receptor. Loss of npr-4 and npr-5 did not induce ascr#8 avoidance; by contrast, KO of npr-10 showed avoidance to ascr#8, although the avoidance was not as robust as in the flp-3 KOs, suggesting other FLP-3 receptors are aiding in the suppression response. Using CRISPR/Cas, the authors generated a frpr-16 KO, which also showed avoidance to ascr#8, although again, not as robustly as the flp-3 KO. A double npr-10; frpr-16 double knockout, surprisingly, showed no avoidance response; this result is particularly confounding as the double KOs had a decreased dwell time. This discrepancy should be addressed further in the Discussion.

To confirm that FLP-3 peptides could activate NPR-10 and FRPR-16, the npr-10 and frpr-16 constructs were transfected into CHO cells and different FLP-3 peptides applied. With the exception of FLP-3-6 and FLP-3-10, the remaining FLP-3 peptides could activate the receptors at nM concentrations; FRPR-16 could be activated at lower FLP-3 concentrations than NPR-10. However, these results are somewhat different than what was found in the peptide rescue experiments, but the discrepancy was not addressed in the Discussion.

Lastly, the authors used an innovative method to determine whether the chemoattractive response to ascr#8 could be rescued by feeding bacteria that contained different FLP-3 constructs. However, it is somewhat confusing from these data (and the referenced paper) how the peptides are generated. Bacteria are transformed with the flp-3 pGEX constructs, which were not described in Materials and Methods and described cursorily in the legend. The bacteria presumably generate the propeptide, but the construct does not have a signal peptide and it is unclear how the protein gets properly modified (cleaved and amidated) and released. Are the authors assuming that intestinal digestion of bacteria release the propeptides for modification by *C. elegans*? For a new technique, there should be more

details and indication that the peptides are made (e.g., do bacterial lysates bind an anti-FLP antibody on a dot blot?). The FLP-3 constructs have a His tag, but it is unclear how the His tag was used. While the data looked promising and potentially a novel method of applying neuropeptides, more controls and explanations need to be included. The conclusions that only specific FLP-3 neuropeptides could rescue assumes that all neuropeptides have equal turnover and degradation rates. The N-terminal of each FLP-3 peptide differs slightly and could affect its protection from proteases, turnover rate, and/or degradation rates, thereby leading to lack of rescue.

The legends are all somewhat skimpy; additional details could be added to all. Legends that are particularly lacking are indicated below. Overall, the experiments are very cleverly designed and provide an excellent basis for understanding sex-specific behaviors. The data are well-presented but some need further clarifications.

Minor comments:

1. The differences between the S (spatial) and V (vehicle=water) controls were not explicitly delineated. Both S and V were used as controls and should presumably have the same values, yet in many instances were significantly different from each other. The vehicle control added liquid to the spot, whereas no liquid was added to the spatial controls, so it is unclear why there were differences between the two. For instance, why did *osm-3* male mutants and *him-5* hermaphrodite mutants dwell significantly longer on the vehicle than the spatial control (Figs. 1B and 1D)?

2. Fig. 2 legend: A bit too concise. Should be fleshed out more to understand what was done.

3. Table in 3A: The Li group cited values incorrectly. The values should be microM, not mM, for C16D6.2 and Y58G8A.4.

4. Lines 193-196: The authors assume that all FLP-3 peptides have the same function. However, different FLP-3 peptides could have different functions, some of which are not related to the pheromone response.

5. Supp. Fig. 5 legend: The legend is confusing. Is this figure just showing data for *ascr#3*? If so, remove mention of *ascr#8* from the legend.

6. Fig. 6 legend: As with Materials & Methods, there needs to be more explanation.

Response to Reviewers' comments:
Reviewer #1 (Remarks to the Author):

This is a very interesting paper by Reilly et al. where they find that several peptides encoded by the flp-3 gene determine the valence of the pheromone component ascr#8 in C. elegans males. One major strength of the work is the breadth of approaches they employ. They use a combination of new behavioural assays, genetics, gene expression analysis, genome editing, biochemistry and rescue-by-feeding to show that the peptides flp-3.2 and 3.9 act through their receptors npr-10 and frpr-16 to switch the behaviour of males towards ascr#8 from repulsion to attraction. In addition, they authors are able to dissect which peptides and perhaps even amino acids are important for a two-step process of suppression of avoidance and promotion of attraction. How sensory perception and valence of stimuli are encoded in the nervous system at the molecular and cellular level is a major question in neurobiology and one that this journal's broad readership will be interested in.

Major comments:

Reviewer Comment-1: The authors refer to supplemental figure 2B to state that only a small percentage of animals (30-45%) exhibit attractive visits to the cue but the graph shows % of attractive visits per worm not % of worms visiting the spot. Where is the data showing % animals? And also, what is the % of animals showing avoidance? This data seems important because the authors state that the CEMs are the only sensors of ascr#8 (for both attraction and avoidance) and that the 35-40% responses matches well the rate of CEM calcium transients upon ascaroside exposure.

Author Response-1: We appreciate the reviewer's critical analysis of our data and interpretation. In Supp. Figure 2B, we show that males exhibit 30-45% attractive visits to ascr#8. As the reviewer points out, this is calculated as the percentage of attractive visits per worm, not the percentage of animals visiting the spot.

We have reanalyzed our data sets to investigate the percent of worms that attracted to the spot (below, Supp. Fig 2C,F). Similar to the percent attractive visits per worm, all wildtype strains show a higher percentage of attraction to ascr#8 over the vehicle control.

The Avoidance Index (AI) data displayed in the main text and figures is a direct correlate of the % of animals exhibited avoidance to ascr#8. The AI is calculated as the number of animals avoiding ascr#8 divided by the total number of animals tested. As such, the AI is merely the percent of animals avoiding divided by 100.

Supp. Fig 2C: Percentage of male worms that exhibited any attraction to *ascr#8*. Dark grey denotes vehicle controls (“V”), colors denote *ascr#8* values (“A”) (N2, blue; *him-5*, red; *him-8*, purple; *osm-3;him-5*, orange). Error bars denote SEM. $n \geq 5$. * $p < 0.05$, ** $p < 0.01$, *** $p < 0.001$, **** $p < 0.0001$. Paired *t*-test of V vs. A.

Reviewer Comment-2: The authors conclude that the CEM neurons are the only source of ascr#8 chemosensation based on the fact that ceh-30 mutants (which lack CEMs) are neither attracted to nor repelled by ascr#8. This conclusion is an overstatement. The experiment shows that the CEMs are indeed required for sensation, however, it doesn't exclude the possibility that other neurons may also be required (despite not being sufficient to drive behavioural responses in the absence of the CEMs). Furthermore, because npr-10, one of the receptors for flp-3 which is important for the behavioural responses to ascr#8, is expressed in ADL, a neuron shown to sense other ascarosides in a sexually dimorphic manner, the authors should test the requirement of ADL in ascr#8 sensation.

Author Response-2: We thank the reviewer for pointing out the interpretation of results to our *ceh-30 lof* experiments. We agree that other neurons may be indirectly involved and have changed the wording of the results (lines 231-232) to refrain from excluding the possibility of the involvement of other neurons, as they may be unable to elicit behavioral responses on their own.

Original text: “confirming that the CEM neurons are the sole source of *ascr#8* chemosensation in male *C. elegans*.”

Changed text: “confirming that the CEM neurons are the primary route of *ascr#8* chemosensation which results in the male *C. elegans* behavioral response to *ascr#8*.”

However, we argue that testing the involvement of the ADL sensory neuron will not provide an additional meaningful outcome within the scope of this paper. We have shown that (1) loss of the CEM neurons (*ceh-30*) abolishes the ability of males to respond attractively to *ascr#8*, and that (2) a *ceh-30;flp-3* double mutant still does not avoid the pheromone.

While ADL does express NPR-10, a receptor we propose serves to propagate the *flp-3* signal, ablation of the ADL neurons (and therefore NPR-10) would not add to our understanding of regulating the avoidance of *ascr#8*, as *ceh-30;flp-3* animals already do not avoid. We feel that additional inquiry into the involvement of ADL in *ascr#8* sensation, while beneficial in understanding sexually dimorphic pheromone sensation, will not add to this manuscript's insights regarding *flp-3* gene's regulation of innate behaviors.

Reviewer Comment-3: Through the rescue-by-feeding experiments, the authors very nicely show that some peptides specifically drive attraction and others suppress avoidance. Based on the sequence of such peptides, they propose that a Threonine in the 9th position from the C terminus of the peptide is critical for suppression. This conclusion could be further strengthened with a direct manipulation: by testing whether switching the E in that position by a T in *flp-3.7* and feeding it, rescues the avoidance phenotype of *flp-3* mutants.

Author Response-3: We thank the reviewer for suggesting this vital experiment, to verify the role of the Threonine in mediating avoidance to the small-molecule *ascr#8*. We generated the variants using site-directed mutagenesis and fed these variants to the *flp-3 lof* worms. We discovered that while FLP-3-7 does not rescue either attraction or avoidance behavior, FLP-3-7T (the same peptide, with a Threonine replacing a Glutamate) is indeed able to suppress the avoidance behavior. These data are depicted in panels Figure 6 E, F, G, H.

Simultaneously, these experiments serve to lend credence to our second hypothesis that the terminal NPEND sequence of FLP-3-9 is what drives attraction: FLP-3-7 contains amino acids prior to the threonine and is unable to drive attraction.

These data have been added as **Figure 6 E-H**, and **Supp. Figure 13**. Similarly, the text has been updated in all relevant sections.

Figure 6E-H: (E) Mutational schematic of FLP-3-7 to generate FLP-3-7T: the glutamate (“E”) in position 9 was mutated to a threonine (“T”). (F) Raw dwell time and (G) log(fold-change) of *flp-3* animals raised on FLP-3-7 and FLP-3-7T displays no rescue of behavior. (H) Avoidance indexes of *flp-3* animals raised on FLP-3-7 and FLP-3-7T revealed a suppression of avoidance behavior only in the presence of the mutated peptide.

*Reviewer Comment-4: Lines 301-302, Figure 5E – the *npr-10* and *frpr-16* double mutant has a somewhat surprising phenotype where the avoidance phenotype of single mutants is suppressed. The authors only state that the phenotype is not additive and therefore the receptors are not redundant. However, here or in the discussion, they should provide some potential explanations for the wildtype-like phenotype of the double mutant.*

Author Response-4: We have adjusted the text within the results section to correct this statement. **Original text:** “However, a double mutant containing both *npr-10* and *frpr-16* null alleles did not result in an additive effect in the avoidance phenotype, suggesting that these receptors are non-redundant in their functions (Figure 5E, Supp. Figure 11C).”

Changed Text: “Interestingly, a double mutant containing both *npr-10* and *frpr-16* null alleles not only did not result in an additive effect in the avoidance phenotype, but in fact suppressed the avoidance phenotype, suggesting that these receptors are non-redundant in their functions (Figure 5E, Supp. Figure 11C).”

In the discussion section, we have also elaborated further on this, as while one mutant results in a skewed behavioral response, loss of both abolishes the behavioral response:

New text: “We argue that is the loss of both of these modules that underlies the lack of *ascr#8* behavioral response in *npr-10;frpr-16* double mutant animals (**Figure 3, 5**).”

Minor comments:

*Reviewer Comment-5: When they describe the expression pattern of *flp-3* for the first time (p11 and 12) it is not clear whether IL1 and SPD are the only sites of expression and whether this is male-specific expression or whether hermaphrodites also have expression in IL1.*

Author Response-5: We apologize for the lack of clarity in our description of the *flp-3* expression pattern. Within the male *C. elegans*, IL1 and SPD are the only sites of expression – which is now clarified in the text. Characterizing the expression pattern of the translational fusion of *flp-3* in hermaphrodites indicates that *flp-3* expressed in the amphid IL1 neurons. Comparing the expression patterns in the two sexes suggests that *flp-3* expression in the male-specific neuron SPD making *flp-3* tail expression sex-specific.

The revised version of the manuscript now includes an image depicting hermaphrodite *flp-3* localization within IL1 neurons, alongside the supplementary figure depicting the same for the male (**Supp. Figure 6**). We have also updated the text describing the expression pattern of *flp-3* to reflect this clearer finding (Lines 216-226).

Supp. Figure 6 E, F: (E, F) IL1 expression of *pflp-3::flp-3::mCherry*. mCherry is faintly observed in the IL1 soma (arrows). The fluorescent protein is also observed in the dendritic cilia of the IL1 neurons (bars), as well as in punctate vesicles along the dendrites (arrowheads).

Reviewer Comment-6: The figures need to be better labelled stating whether the data refers to males or hermaphrodites; to ascr-8 or ascr-3; the expression patterns are of the ligand or the receptors; etc. Otherwise, it gets a bit confusing.

Author Response-6: We thank the reviewer for pointing out this lack of clarity in the expression image panels. We have now added male and hermaphrodite symbols to all figure panels depicting data (schematic panels do not include symbols). We have also edited the text of Supp. Figure 5 denoting that the data is relating to *ascr#3*, not *ascr#8*, for that figure only.

Reviewer Comment-7: Line 212 – PQR is a sensory neuron, not an interneuron

Author Response-7: We appreciate the reviewer noting this oversight and have since updated the text to correct this mistake.

Reviewer #2 (Remarks to the Author):

*Reviewer Comment-1: Males and hermaphrodites have differential responses to several pheromones in the nematode *C. elegans*. Males have an attractive response to the pheromone ascaroside #8 (*ascr#8*), while hermaphrodites show avoidance. Neuropeptides, such as FLP-3, 6, 12, and 19, are expressed in several male, sex-specific neurons. The authors premise that some pheromone responses are mediated through neuropeptide signaling. Response to *ascr#8* was examined in this paper.*

*Animals were tested for their dwell time on and attraction to *ascr#8*. Wild-type males dwell on the *ascr#8* spot for significantly more time than hermaphrodites, which avoid *ascr#8*. *flp-6*, 12, and 19 male mutants showed similar dwell times as wild-type males. By contrast, *flp-3* male mutants showed significantly lower dwell times on and avoided *ascr#8*, suggesting that FLP-3 peptides act to suppress *ascr#8* avoidance. The response of *flp-3* mutants to *ascr#3*, a different chemoattractive pheromone, was unaffected, nicely showing that responses to *ascr#8* and #3 are mediated by different pathways.*

*CEH-30 transcription factor is necessary for specification of the male-specific CEM neurons. Knockout (KO) of the CEMs in males did not result in an avoidance response to *ascr#8* in a wild-type or *flp-3* mutant background, suggesting that FLP-3 peptides are released from CEMs to suppress the avoidance response.*

Author Response-1: We apologize for the lack of clarity in the text describing CEM neurons. We do not believe that the CEM neurons are responsible for releasing FLP-3 peptides. Based on our expression pattern analyses, the *flp-3* translational fusion does not express within CEM neurons, but instead the IL1 neurons of the amphid region. (Figure number)

We instead argue that the CEM neurons are a major component of the *ascr#8* sensory network, and that this sensory network is upstream of a behavioral network that is modulated by FLP-3 signaling.

Original text: “confirming that the CEM neurons are the sole source of *ascr#8* chemosensation in male *C. elegans*.”

Changed text: “confirming that the CEM neurons are the primary route of *ascr#8* chemosensation which results in the male *C. elegans* behavioral response.”

*Reviewer Comment-2: Several FLP-3 receptors have been previously identified: NPR-4, NPR-5, and NPR-10; the authors also classify the *Drosophila* homologue FRPR-16 as a FLP-3 receptor. Loss of *npr-4* and *npr-5* did not induce *ascr#8* avoidance; by contrast, KO of *npr-10* showed avoidance to *ascr#8*, although the avoidance was not as robust as in the *flp-3* KOs, suggesting other FLP-3 receptors are aiding in the suppression response. Using CRISPR/Cas, the authors generated a *frpr-16* KO, which also showed avoidance to *ascr#8*, although again, not as robustly as the *flp-3* KO. A double *npr-10; frpr-16* double knockout, surprisingly, showed no avoidance response; this result is particularly confounding as the double KOs had a decreased dwell time. This discrepancy should be addressed further in the Discussion.*

Author Response-2: We agree with the reviewer's suggestion. In our revised version, we have elaborated our explanation as to how NPR module loss affects the behavioral response on further in the Discussion. (Lines 412-217).

New text: "We argue that it is the loss of both of these modules that underlies the lack of ascr#8 behavioral response in *npr-10;frpr-16* double mutant animals (Figure 3, 5); the loss of one module results in a skewed behavioral response, while the loss of both NPR-10 and FRPR-16 modules abolishes the ability of the animal to respond to ascr#8. Future studies incorporating cell-specific rescue of both NPR-10 and FRPR-16 will further elucidate this circuitry."

Reviewer Comment-3: To confirm that FLP-3 peptides could activate NPR-10 and FRPR-16, the npr-10 and frpr-16 constructs were transfected into CHO cells and different FLP-3 peptides applied. With the exception of FLP-3-6 and FLP-3-10, the remaining FLP-3 peptides could activate the receptors at nM concentrations; FRPR-16 could be activated at lower FLP-3 concentrations than NPR-10. However, these results are somewhat different than what was found in the peptide rescue experiments, but the discrepancy was not addressed in the Discussion.

Author Response-3: We appreciate the Reviewer's comment on our biochemical data and its relevant conclusions. We would like to clarify our interpretation of the in-vitro binding efficiency of the receptor to the ligand and we believe that this binding efficiency does not require perfect correlation to *in vivo* potencies.

In the heterologous system employed in our manuscript as well as other manuscripts that have used this technology (Nelson et al., PLoS One, 2015; Iannacone et al., Elife, 2017, Van Sinay et al., PNAS, 2017; Peymen et al, PLoS Genetics, 2019), a worm receptor is expressed in mammalian cells. These cells inherently may differ in their innate G protein coupling systems, co-factors, and even the functionality of the receptor. To obtain EC₅₀ values for the most receptors, the human G alpha 16 protein is used.

However, while we agree that *in vivo*, these receptors may couple with other G proteins at varying levels of efficiency, the EC₅₀ values obtained in our in-vitro binding studies should be considered as indicative of the physiological relevance of the interaction: nM affinities tend to be functionally relevant.

The behavioral experiments in this study are not a verification of binding affinity, but rather a confirmation of ligand-receptor activity *in vivo*. Though FRPR-16 may exhibit higher affinities than NPR-10 *in vitro*, this may change *in vivo*. Furthermore, the full ascr#8 behavioral circuitry is much more complex than the valence-control mechanism elucidated in this study.

Reviewer Comment-4: Lastly, the authors used an innovative method to determine whether the chemoattractive response to ascr#8 could be rescued by feeding bacteria that contained different FLP-3 constructs. However, it is somewhat confusing from these data (and the referenced paper) how the peptides are generated. Bacteria are transformed with the flp-3 pGEX constructs, which were not described in Materials and Methods and described cursorily in the legend. The bacteria presumably generate the propeptide, but the construct does not have a signal peptide and it is unclear how the protein gets properly modified (cleaved and amidated) and released. Are the authors assuming that intestinal digestion of bacteria release the propeptides for modification by C. elegans? For a new technique, there should be more details and indication that the peptides are made (e.g., do bacterial lysates bind an anti-FLP antibody on a dot blot?). The FLP-3

constructs have a His tag, but it is unclear how the His tag was used. While the data looked promising and potentially a novel method of applying neuropeptides, more controls and explanations need to be included. The conclusions that only specific FLP-3 neuropeptides could rescue assumes that all neuropeptides have equal turnover and degradation rates. The N-terminal of each FLP-3 peptide differs slightly and could affect its protection from proteases, turnover rate, and/or degradation rates, thereby leading to lack of rescue.

Author Response-4: We appreciate the reviewer's comment on the lack of clarity in the manuscript of our novel rescue paradigm. In the revised version, we added more details to provide a clear explanation of the method. To clarify the reviewer's comment, we would like to highlight the some of the important aspects of the technique:

1. We did not use pGEX, but instead generated peptide-encoding plasmids using Gateway Cloning technology: first into a p1-p2 vector, and finally pDEST527: a bacterial expression vector with a T7 promoter which is induced by IPTG.
2. While we agree that it is likely that the bacteria produces the propeptide, we believe that it is the mRNA that is taken up by the nematode for functional translation.
3. The His tag is a component of the expression vector used previously (Xu et al., Peptides, 2017, doi: 10.1016/j.peptides.2017.01.003). We did not alter the vector used in our technique. While we agree that it would be interesting to understand the mechanisms of the feeding protocol in this manuscript, we have now submitted for review a manuscript that provides more detail on the mechanisms underlying the technique DiLoreto, Reilly and Srinivasan (Scientific Reports in review, biorxiv, doi: 10.1101/2021.05.10.443513). Our studies suggest that the mechanism of uptake is via mRNA uptake over direct peptide uptake.

Reviewer Comment-5: The legends are all somewhat skimpy; additional details could be added to all. Legends that are particularly lacking are indicated below. Overall, the experiments are very cleverly designed and provide an excellent basis for understanding sex-specific behaviors. The data are well-presented but some need further clarifications.

Author Response-5: The revised version of the manuscript contains detailed description of each of the figure legends and we thank the reviewer for pointing out the lack of description in our figure captions.

Minor comments:

Reviewer Comment-6: The differences between the S (spatial) and V (vehicle=water) controls were not explicitly delineated. Both S and V were used as controls and should presumably have the same values, yet in many instances were significantly different from each other. The vehicle control added liquid to the spot, whereas no liquid was added to the spatial controls, so it is unclear why there were differences between the two. For instance, why did osm-3 male mutants and him-5 hermaphrodite mutants dwell significantly longer on the vehicle than the spatial control (Figs. 1B and 1D)?

Author Response-6: We thank the reviewer for pointing out our lack of elaboration on the differences between spatial and vehicle controls. We have since added to the text our explanations (Lines 123-126).

Original text: “In this assay, individual animals are placed directly into the spot of the ascaroside cue while simultaneously removing any potential of male-male contact.”

Revised text: “In this assay, individual animals are placed directly into the spot of the ascaroside cue while simultaneously removing any potential of male-male contact. A spatial control is included throughout the assay plate to allow us to investigate any innate differences in the number of visits to the well center, or the time spent therein, of which we have only found one strain to date with differences in male dwell time (Figure 1B).”

In summary, some mutant strains exhibit ‘edge effects’, we wanted to be able to spot differences in how often strains visited the center of the well with no controls (spatial), and what affect water (vehicle) had on that. *osm-3* mutant males (**Figure 1B**) is the only strain that statistically exhibited an increase in vehicle dwell time over the spatial control.

Figure 1D displays hermaphrodite data. Hermaphrodites repeatedly exhibit differences in spatial vs. vehicle dwell time (which is abolished in *ascr#8* conditions). But as this study is focused on investigating male attraction to *ascr#8*, we have not determined the mechanisms driving the increase in hermaphrodite vehicle dwell time.

Reviewer Comment-7: Figure 2 legend: A bit too concise. Should be fleshed out more to understand what was done.

Author Response-7: The revised manuscript expands the description of this figure legend to add more clarity and aid the reader in more fully understanding what was done in these experiments.

Reviewer Comment-8: Table in 3A: The Li group cited values incorrectly. The values should be microM, not mM, for C16D6.2 and Y58G8A.4.

Author Response-8: We thank the reviewer for bringing this inconsistency to our attention. We have since corrected the values for C16D6.2 and Y58G8A.4 to reflect the correct micromolar affinities in the revised manuscript.

Reviewer Comment-9: Lines 193-196: The authors assume that all FLP-3 peptides have the same function. However, different FLP-3 peptides could have different functions, some of which are not related to the pheromone response.

Author Response-9: We agree with the reviewer that the other peptides can have functions not related to pheromone responses. Our data clearly shows that FLP-3-2 suppresses the avoidance response, while FLP-3-9 drives attraction (Figure 6). Since the other peptides did not affect *ascr#8* responses, we interpreted that these peptides do not play a role in pheromone responses at all. We have added textual clarification throughout the revised text to imply the same.

*Reviewer Comment-10: Supp. Figure 5 legend: The legend is confusing. Is this figure just showing data for *ascr#3*? If so, remove mention of *ascr#8* from the legend.*

Author Response-10: We have changed the title of the figure legend to read, “Loss of *flp-3* does not affect male behavioral response to *acsr#3*.” We have additionally added text over each panel clarifying that this figure is depicting data for *acsr#3*.

Reviewer Comment-11: Figure 6 legend: As with Materials & Methods, there needs to be more explanation.

Author Response-11: We have added description to include more detail to the figure legend in the revised version of the manuscript.

Old text: “Peptide feeding rescues wild-type behavior and reveals two active peptides within the FLP-3 precursor. (A) Overview of rescue-by-feeding paradigm. **(Top)** the peptide of interest is flanked by EGL-3 cleavage sites, with a 6x-His tag upstream. **(Bottom)** *flp-3* lof animals are raised on bacteria expressing a FLP-3 peptide of interest and are assayed as young adults. **(B)** Avoidance indexes of *him-8* and *flp-3* animals raised on scramble, FLP-3-1, FLP-3-2, FLP-3-4, FLP-3-9, and FLP-3-10 peptides. **(C)** Raw dwell time and **(D)** log(fold-change) values for *him-8* and *flp-3* animals raised on scramble, FLP-3-2, and FLP-3-9 peptides. Active peptides shown in pink.”

Revised text: “Peptide feeding rescues wild-type behavior and reveals two active peptides within the FLP-3 precursor. (A) Overview of rescue-by-feeding paradigm: **(Top)** A plasmid is generated which encodes the peptide of interest is flanked by EGL-3 cleavage sites, with a 6x-His tag upstream. **(Bottom)** *flp-3* lof animals are raised on bacteria expressing the FLP-3 peptide of interest and are assayed as young adults. **(B)** Avoidance indexes of *him-8* and *flp-3* animals raised on scramble peptide or specified FLP-3 peptides. **(C)** Raw dwell time and **(D)** log(fold-change) values for *him-8* and *flp-3* animals raised on scramble peptide and FLP-3 peptides which suppressed avoidance (FLP-3-2 and FLP-3-9) peptides. **(E)** Mutational schematic of FLP-3-7 to generate FLP-3-7T: the glutamate (“E”) in position 9 was mutated to a threonine (“T”). **(F)** Raw dwell time and **(G)** log(fold-change) of *flp-3* animals raised on FLP-3-7 and FLP-3-7T displays no rescue of behavior. **(H)** Avoidance indexes of *flp-3* animals raised on FLP-3-7 and FLP-3-7T revealed a suppression of avoidance behavior only in the presence of the mutated peptide.”

Reviewers' comments:

Reviewer #1 (Remarks to the Author):

The authors have satisfactorily addressed my initial queries and suggestions. I am very pleased to see that the experiment I suggested on mutating one of the peptides was successful at strengthening several of their arguments and conclusions.

Reviewer #2 (Remarks to the Author):

This paper examines the sex-specific response to a pheromone, *ascr#8*, in *C. elegans*. Hermaphrodites avoid *ascr#8*, whereas males do not. After testing different flp genes that are expressed in male-specific neurons, the authors present a compelling case that signaling through the FLP-3 neuropeptide mediates the lack of repulsion in males. Without activation of a male-specific circuit that includes flp-3 expression, male animals are also repulsed by *ascr#8*.

In this revised manuscript, the authors addressed many of the previous issues. However, several points should still be addressed in the Discussion.

1. Why do *him-8* males dwell on *ascr#8* so much longer than *him-5* males (Supp. Fig. 1), but a smaller percentage are attracted to *ascr#8* than *him-5* males (Supp. Fig. 2)?
2. Although the focus of this paper is on male and not hermaphrodite responses, nevertheless, why does the V values differ so much from the S values in hermaphrodites? What do these differences imply?
3. The authors present data to indicate that male-specific CEMs are involved in the *ascr#8* response, but they are not included in Fig. 8. The authors equivocate on how CEMs are involved, but should present a model for its action in Fig. 8.
4. The authors only test peptides singly; however, different FLP-3 peptides may synergize with FLP-3-2 and/or FLP-3-9 to elicit a greater response. This possibility could be explored in the Discussion.
5. The assumption is that if there is no rescue with a specific FLP-3 construct, that implies that the peptide has no function in the *ascr#8* response. However, there could be many reasons why a response was not seen. For instance, the peptide RNA could be unstable or rapidly degraded, so none of the peptide was made. While the positive results are indicative of rescue, negative results should be considered more carefully and caveats should be included in the Discussion.
5. Loss of *npr-10* or *frpr-16* results in aversion to *ascr#8*. But if both genes are deleted, no aversion is seen. The explanation in Discussion is un-focused and hard to understand. The gist (I think) is that FLP-3/FRPR-16 and FLP-3/NPR-10 signaling in non-sex specific cells mediates *ascr#8* aversion in males and hermaphrodites. FLP-3/NPR-10 signaling in male-specific cells suppresses this *ascr#8* aversion. Whatever the model, the Discussion needs to be re-written and clarified so that it is more coherent.

Overall, this paper presents an interesting mechanism by which different sexes respond to a pheromone; in particular, it shows how both the common and sex-specific neural circuitry is used to elicit a different response in the sexes. This work has particular relevance as FLP-3 homologues are present in parasitic nematodes. In addition, a new method to test peptides was developed; such a method would be very useful for other peptide studies.

Some minor corrections:

1. line 194: missing "way" in sentence.
2. Should be μM in Fig. 3A table.
3. line 273: not a full sentence.
4. lines 315, 320: Supp. Fig. 4 is the incorrect citation. Perhaps it should be Supp. Fig. 10? The presentation of peptides was very confusingly presented in this section and needs to be re-written. It is the first presentation of the peptide data, but it is written assuming that the reader already knows which peptides are responsible for suppressing the aversion.

Reviewers' comments:

Reviewer #1 (Remarks to the Author):

Reviewer Comment-1: The authors have satisfactorily addressed my initial queries and suggestions. I am very pleased to see that the experiment I suggested on mutating one of the peptides was successful at strengthening several of their arguments and conclusions.

Author Response: We reiterate our gratitude to the reviewer for their comment on the mutation experiment as we feel it strengthened our arguments on the role of individual amino acid in regulating functional responses. We are glad about the positive response to the revised version of the manuscript.

Reviewer #2 (Remarks to the Author):

Reviewer Comment-2: This paper examines the sex-specific response to a pheromone, ascr#8, in C. elegans. Hermaphrodites avoid ascr#8, whereas males do not. After testing different flp genes that are expressed in male-specific neurons, the authors present a compelling case that signaling through the FLP-3 neuropeptide mediates the lack of repulsion in males. Without activation of a male-specific circuit that includes flp-3 expression, male animals are also repulsed by ascr#8.

In this revised manuscript, the authors addressed many of the previous issues. However, several points should still be addressed in the Discussion.

Reviewer Comment-3: Why do him-8 males dwell on ascr#8 so much longer than him-5 males (Supp. Fig. 1), but a smaller percentage are attracted to ascr#8 than him-5 males (Supp. Fig. 2)?

Author Response: We agree with the reviewer that *him-8* males do spend a significantly more time in *ascr#8* than *him-5* males in **Supp. Fig. 1**. However, we would like to clarify that this data was obtained using the Spot Retention Assay (SRA), and not the Single Worm Attraction Assay (SWAA), which we describe in our current manuscript. We believe that the Spot retention assay (SRA) allows for male-male contact, a feature was skewing our attraction values. This drawback made us develop the novel Single Worm Attraction Assay (SWAA) which enables a better handle on the characterizing the attraction behavior of males in response to pheromones. As depicted in **Fig. 1**, using Single Worm Attraction Assay (SWAA), both *him-5* and *him-8* males spend nearly identical times in *ascr#8*. We propose that while the SRA does allow for characterizing the attractive behavior of *him-5* and *him-8* males, the differences between the two *him* strains shown in **Supp. Fig. 1** is due to male-male contact.

We would like to state that **Supp. Fig. 1** and **Supp. Fig. 2** are displaying data from the two different assays. **Supp Fig. 1** is showing data from SRA and **Supp Fig. 2** is showing data from SWAA. The data presented in Supp. Fig. 2B, which calculates the percent of males attracted should be compared to **Fig. 1** and not **Supp. Fig. 1**. By using this

comparison, we observe that the slight decrease in *him-8* male attraction in **Supp. Fig. 2B** matches the slight, but not statistically significant, decrease in *him-8* male dwell time seen in **Fig. 1 B,C**.

We realize that after initially presenting our attraction data for SWAA in our Results section, we did not discuss the advance in attraction assays in our Discussion. In our revised version, we have added an additional section to the discussion addressing the differences between the two assays (**Lines 374-391**).

Reviewer Comment-4: Although the focus of this paper is on male and not hermaphrodite responses, nevertheless, why does the V values differ so much from the S values in hermaphrodites? What do these differences imply?

Author Response: We fully agree with the reviewer that the results of hermaphrodite response to *ascr#8* in our novel SWAA are not the focus of this manuscript. Though our manuscript focuses on male responses to *ascr#8*, we tested hermaphrodites to observe the robustness of our assay. Our observations suggest that hermaphrodites don't move significantly in our SWAA. These findings corroborate previous studies that demonstrated that hermaphrodites rarely leave food, as they do not need to locate a mate to reproduce; though males will leave food in search of potential mates (Lipton, *et al.*, JNeurosci, 2004).

Previous studies have shown that hermaphrodites prefer the edges of lawns and such "edge effects" cause hermaphrodites not to move from that region (Barrios, *et al.*, Curr. Biol., 2008). However, in the presence of vehicle (V), hermaphrodites display some preference to the area containing the vehicle over what is offered near the edge. While we do not know what causes the hermaphrodites prefer vehicle region, we do not believe that this preference of the region containing the vehicle is different from the Spatial control (S), where they spend more time on the edge. Our data in **Fig. 1** suggests that hermaphrodites avoid *ascr#8*, as observed by hermaphrodites leaving the *ascr#8* drop immediately, and never returning, resulting in low dwell times (**Fig. 1**).

Reviewer Comment-5: The authors present data to indicate that male-specific CEMs are involved in the ascr#8 response, but they are not included in Fig. 8. The authors equivocate on how CEMs are involved, but should present a model for its action in Fig. 8.

Author response: We thank the reviewer for the suggestion to include the male-specific CEM neurons in our schematic. Our original summary figure **Fig. 7**, aimed to display only the FLP-3 regulated circuitry. The role of CEM neurons in sensing male-attracting ascarosides has been previously interrogated by multiple labs, including our own (Chasnov *et al.*, PNAS, 2007, Pungaliya *et al.*, PNAS, 2009, Narayan *et al.*, PNAS, 2016, Reilly *et al.*, JoVE, 2017). However, we agree with the reviewer that since we argue that this is CEM-driven network, that these neurons should be included in this network. The revised manuscript now has an **Fig. 7B** to include the male-specific CEM neurons and their potential synaptic connections (See below). We have since added clarifications of this in the Discussion as well (**Lines 480-483**).

Reviewer Comment-6: The authors only test peptides singly; however, different FLP-3 peptides may synergize with FLP-3-2 and/or FLP-3-9 to elicit a greater response. This possibility could be explored in the Discussion.

Author Response: We agree that these peptides may work in synergistic manners. In fact, given the complexity of the *flp-3* precursor, and the fine-tuned modulation of the *ascr#8* behavioral response, it is likely. We have added to our discussion the likely synergistic roles these peptides play in regulating this behavior (**Lines 453-457**).

*Reviewer Comment-7: The assumption is that if there is no rescue with a specific FLP-3 construct, that implies that the peptide has no function in the *ascr#8* response. However, there could be many reasons why a response was not seen. For instance, the peptide RNA could be unstable or rapidly degraded, so none of the peptide was made. While the positive results are indicative of rescue, negative results should be considered more carefully and caveats should be included in the Discussion.*

Author Response: We thank the reviewer for further insights into how our peptide method functions. We have added a section to our Discussion elaborating on the evaluation of “negative results” in our feeding rescue experiments (**Lines 457-462**).

*Reviewer Comment-8: Loss of *npr-10* or *frpr-16* results in aversion to *ascr#8*. But if both genes are deleted, no aversion is seen. The explanation in Discussion is un-focused and hard to understand. The gist (I think) is that FLP-3/FRPR-16 and FLP-3/NPR-10 signaling in non-sex specific cells mediates *ascr#8* aversion in males and hermaphrodites. FLP-3/NPR-10 signaling in male-specific cells suppresses this *ascr#8* aversion. Whatever the model, the Discussion needs to be re-written and clarified so that it is more coherent.*

Author Response: We apologize for the lack of clarity in the text describing our model. We have rewritten this section of the Discussion in the hopes of clarifying our proposed NP/NPR module modulated circuit (**Lines 436-452**).

Overall, this paper presents an interesting mechanism by which different sexes respond to a pheromone; in particular, it shows how both the common and sex-specific neural circuitry is used to elicit a different response in the sexes. This work has particular relevance as FLP-3 homologues are present in parasitic nematodes. In addition, a new method to test peptides was developed; such a method would be very useful for other peptide studies.

We thank the reviewer for their favorable review and we believe that our revised manuscript presents an interesting mechanism by which different sexes respond to a pheromone and are glad that they see future applications for this work in investigating parasitic nematode biology. We hope that we have appropriately addressed all concerns.

Some minor corrections:

Reviewer Comment-9: line 194: missing “way” in sentence.

Author response: We thank the reviewer for catching this error.

The previous sentence:

“Whether they contribute to the ascr#8 response in an as yet undetermined remains to be determined.”

Now reads:

“Whether they contribute to the ascr#8 response in another way has yet undetermined remains to be elucidated.”

Reviewer Comment-10: Should be μM in Fig. 3A table.

Author response: We thank the reviewer for pointing out this error again. We have uploaded a new Fig. 3A with the correct concentrations noted.

Reviewer Comment-11: line 273: not a full sentence.

Author response: We appreciate the reviewer’s thorough proofreading of our manuscript.

The formerly incomplete sentence:

“Interestingly, the hermaphrodite tail expression in the dorso-rectal ganglion neurons DVA, DVB, DVC, and ALN (**Supp. Fig. 9E**).”

now reads:

“Interestingly, the hermaphrodite tail exhibits expression in the dorso-rectal ganglion neurons DVA, DVB, DVC, and ALN (**Supp. Fig. 9E**), which is not observed in the male tail, suggesting that NPR-10 plays sex-specific roles based on sex-specific expression patterns.”

Reviewer Comment-12: lines 315, 320: Supp. Fig. 4 is the incorrect citation. Perhaps it should be Supp. Fig. 10? The presentation of peptides was very confusingly presented in this section and needs to be re-written. It is the first presentation of the peptide data, but

it is written assuming that the reader already knows which peptides are responsible for suppressing the aversion.

Author response: We thank the reviewer for pointing out the confusing nature of this section of our results. The revised version of the manuscript has an extensively rewritten section, that clarifies which peptides are responsible for suppressing the aversion and presenting the peptide data in an understandable way.

We wish to state that **Supp. Fig. 4** is still the correct citation, as this figure presents the *flp-3(ok3625)* behavioral data, which determines that FLP-3-1 and FLP-3-4 are insufficient to drive the proper behavioral response *in vivo*. **Supp. Fig. 10** is *in vitro* activation data showing how individual peptides activate the NPR-10 and FRPR-16 receptors. In addition, our revised manuscript, we have added a new figure **Fig. 7A** (see below) that provides a schematic overview of the different peptides that display rescue of behavior.

REVIEWERS' COMMENTS:

Reviewer #2 (Remarks to the Author):

The authors have addressed all concerns satisfactorily. The paper presents an interesting way in which an animal uses sex-specific and sex-common neural circuitry to generate different responses to a pheromone. There are only a few very minor edits that would clarify the text.

Minor edits

1. lines 127, 132: Insert "male" before strains.
2. line 194: insert "way" before remains.
3. line 216: Why would dense core vesicles appear in dendrites? Are the authors suggesting that FLP-3 peptides are released from dendrites? Please clarify.
4. line 273: something is missing in the sentence.
5. line 354: "at least" two peptides would be more appropriate, as it is unclear whether other RNA constructs are unstable, thereby leading to no peptide expression.